# SPINK1-induced tumor plasticity provides a therapeutic window for chemotherapy in hepatocellular carcinoma

Ki-Fong Man[1,11], Lei Zhou[2,3,11], Huajian Yu[1], Ka-Hei Lam[1], Wei Cheng [4], Jun Yu [5], Terence K. Lee [6], Jing-Ping Yun [7], Xin-Yuan Guan [2,8], Ming Liu [4,9] & Stephanie Ma [1,2,10] ✉

Tumor lineage plasticity, considered a hallmark of cancer, denotes the phenomenon in which tumor cells co-opt developmental pathways to attain cellular plasticity, enabling them to evade targeted therapeutic interventions. However, the underlying molecular events remain largely elusive. Our recent study identified CD133/Prom1 in hepatocellular carcinoma (HCC) tumors to mark proliferative tumor-propagating cells with cancer stem cell-like properties, that follow a dedifferentiation trajectory towards a more embryonic state. Here we show SPINK1 to strongly associate with CD133 + HCC, and tumor dedifferentiation. Enhanced transcriptional activity of SPINK1 is mediated by promoter binding of ELF3, which like CD133, is found to increase following 5-FU and cisplatin treatment; while targeted depletion of CD133 will reduce both ELF3 and SPINK1. Functionally, SPINK1 overexpression promotes tumor initiation, self-renewal, and chemoresistance by driving a deregulated EGFR-ERK-CDK4/6-E2F2 signaling axis to induce dedifferentiation of HCC cells into their ancestral lineages. Depleting SPINK1 function by neutralizing antibody treatment or in vivo lentivirus-mediated *Spink1* knockdown dampens HCC cancer growth and their ability to resist chemotherapy. Targeting oncofetal SPINK1 may represent a promising therapeutic option for HCC treatment.

The "Hallmarks of Cancer" published by Hanahan and Weinberg provides the original and updated conceptual framework of cancer cell biology[1,2]. In 2022, Hanahan has further published an updated version that highlights additional emerging hallmarks and enabling characteristics, including those under the umbrella of 'unlocking phenotypic plasticity'[3]. Phenotypic plasticity is an acquired hallmark of cancer that enables various disruptions of cellular differentiation, including "dedifferentiation from mature to progenitor states",

[1]School of Biomedical Sciences, Li Ka Shing Faculty of Medicine, The University of Hong Kong, Hong Kong, China. [2]Department of Clinical Oncology, Shenzhen Key Laboratory for Cancer Metastasis and Personalized Therapy, The University of Hong Kong – Shenzhen Hospital, Hong Kong, China. [3]Precision Medicine Institute, The First Affiliated Hospital, Sun Yat-Sen University, Guangzhou, China. [4]Guangzhou Municipal and Guangdong Provincial Key Laboratory of Protein Modification and Degradation, School of Basic Medical Science, Guangzhou Medical University, Guangzhou, China. [5]Institute of Digestive Disease and The Department of Medicine and Therapeutics, State Key Laboratory of Digestive Disease, Li Ka Shing Institute of Health Sciences, The Chinese University of Hong Kong, Hong Kong, China. [6]Department of Applied Biology and Chemical Technology, The Hong Kong Polytechnic University, Hong Kong, China. [7]Department of Pathology, Sun Yat-Sen University Cancer Centre, Guangzhou, China. [8]Department of Clinical Oncology, School of Clinical Medicine, Li Ka Shing Faculty of Medicine, The University of Hong Kong, Hong Kong, China. [9]Affiliated Cancer Hospital and Institute of Guangzhou Medical University, Guangzhou, China. [10]State Key Laboratory of Liver Research, The University of Hong Kong, Hong Kong, China. [11]These authors contributed equally: Ki-Fong Man, Lei Zhou. ✉e-mail: stefma@hku.hk

"blocked differentiation from progenitor cell states" and "transdifferentiation into different cell lineages"[3].

There is now ample evidence to show that hepatocellular carcinoma (HCC) tumors contain less differentiated cells that are resistant to therapy and associated with the development of tumor relapse. Markers of these tumor-initiating/propagating cell subsets have now been extensively identified, with CD133 being one of the best-studied functional and phenotypic markers[4,5]. Our previous work also demonstrated CD133+ cells to be more resistant to chemotherapy, like 5-fluorouracil (5-FU) and cisplatin, as compared to CD133- counterparts[6,7]. Recently, we have also been able to lineage trace a population of CD133/Prominin1 (Prom1)-derived proliferative tumor-propagating HCC cells in vivo[8]. These cells display a trajectory of dedifferentiation towards more embryonic-like and epithelial mesenchymal transition (EMT) features; while targeted depletion of the CD133/Prom1-lineage resulted in the reviving of hepatic functional genes in the tumoral livers[8]. Consistently, ectopic expression of Sox9, a marker of liver progenitor cells, induces Krt19 expression and intrahepatic cholangiocarcinoma-like tumor phenotype, a more aggressive trait of HCC, in Myc+NrasG12V-driven HCC[9]. This collectively suggests that stem/progenitor-like cancer cells play critical roles in maintaining cancer cell plasticity, and depleting these cells or impelling them to differentiate may bring about innovative therapeutic strategies. Unfortunately, CD133 is not specific to HCC but is also expressed in the normal regenerating liver. Identifying critical factors expressed specifically in liver tumor-initiating/propagating CD133 + , but not in CD133+ cells of the regenerating liver, may offer important therapeutic opportunities for overcoming chemoresistance in HCC.

SPINK1, which has previously been found to be overexpressed in HBV and HCV-related HCC and associated with different cancer types, has been reported to promote cell proliferation, increase metastatic and invasive potential, and hold promise as a potential diagnostic marker[10–16]. Despite this, its role in mediating tumor plasticity and chemoresistance as well as its potential therapeutic targeting in HCC has never been explored and is the main focus of this study.

Herein, we show lineage ablation of CD133 cells to sensitize HCC tumors to chemotherapy 5-FU and cisplatin. Through comparing 'HCC' CD133+ and 'normal' CD133+ cells, we identify SPINK1 to be preferentially enriched in CD133 + HCC and to correlate with a poorly differentiated phenotype in mouse/human developing livers as well as aggressive cancer features in HCC clinical samples. Upon 5-FU and cisplatin treatment, ELF3-mediated transcription and secretion of SPINK1 increase, and potentiate HCC cells' ability to promote tumor initiation, stemness, dedifferentiation and chemoresistance, by binding to EGFR and consequently driving the ERK-CDK4/6-E2F2 signaling cascade. Depleting SPINK1 function by neutralizing antibody treatment or in vivo lentivirus-mediated Spink1 knockdown dampens HCC cancer growth and their ability to resist chemotherapy. In summary, SPINK1-induced tumor lineage plasticity may represent the Achilles' heel for surviving chemotherapy-enriched tumor-initiating/propagating cells in HCC. Targeting SPINK1 may widen the therapeutic window for combating chemoresistant cancer stemness in the clinic.

## Results

### 5-fluorouracil and cisplatin chemotherapy enrich for a CD133/Prom1+ liver tumor-initiating/propagating subset in HCC

The use of 5-FU and cisplatin chemotherapies, often administered through transarterial chemoembolization[17] and hepatic artery infusion chemotherapy[18,19], are commonly adopted in clinical practice for management of intermediate/advanced HCC patients. Unfortunately, these treatments are usually followed by chemoresistance, in part a result of the heterogeneity of HCC tumors which we now know can be contributed by the presence of tumor-initiating/propagating cells[20].

Given the frequent activation of both RAS/MAPK and PI3K/AKT/mTOR pathways in almost 50% of HCC patients[21], to examine whether chemotherapy would enrich for CD133+ tumor-initiating/propagating cells in HCC, flow cytometry analysis for CD133 expression was carried out on a hydrodynamic tail vein injected (HTVI) NRasV12+Myr-AKT proto-oncogenes-driven HCC mouse model that was treated either with DMSO control or 5-FU or cisplatin. While 5-FU or cisplatin enriched for a CD133/Prom1+ liver tumor-initiating/propagating cell subset (Fig. 1a and Supplementary Fig. S1a), the genetic depletion of Prom1+ cells, using conditional CreER-induced diphtheria toxin (DTA) crossed with Prom1[C-L], resulting in selective Prom1 cell death (Prom1-DTA), sensitized HCC tumors to 5-FU treatment and impeded tumor growth as demonstrated by reduced liver weight (Fig. 1c, d) and number of tumor nodules (Fig. 1e, f), concomitant with a decreased tumor-initiating cell frequency (Fig. 1g). Importantly, where 5-FU treatment was administered as a standalone therapy (Fig. 1b), it did not demonstrate a reduction in tumor growth (Fig. 1c–f). In fact, it even resulted in an elevated stem cell frequency (Fig. 1g), which raises concerns about the potential for increased tumor recurrence if chemotherapy alone is utilized as a treatment strategy.

### Preferential upregulation of *Spink1* in CD133/Prom1+ tumor-initiating/propagating cells and HCC tumors

While CD133/Prom1 has been demonstrated to have a functional role in HCC by us and others, it is important to note that CD133 is not specific to HCC and is also expressed in the regenerating liver. It is essential to identify altered and targetable molecular players that are specific to CD133+ liver cancer cells to better design drugs that can precisely interfere with tumor-initiating/propagating cells in HCC but not normal stem cell function. We compared the transcriptomic profiles of sorted CD133+ and CD133- cells harvested from regenerating liver (induced by 0.1% DDC diet), HTVI NRasV12+Myr-AKT proto-oncogenes-driven HCC tumors and inflammation-associated DEN +CCl$_4$ HCC tumors (Fig. 2a)[22]. 615 and 701 genes were found commonly upregulated and downregulated, respectively, in the CD133+ cells of the two HCC models but not the liver regenerating model. Upon further correlation with survival using human clinical samples extracted from TCGA-LIHC, *IGFALS*, *SPINK1* and *B4GALNT1* were identified as potential candidates (Fig. 2a, Supplementary Fig. S2a, b). On further analysis, only *SPINK1* demonstrated a similar decrease in expression upon Prom1+ cell depletion in the two HCC mouse models and thus was chosen for studies (Supplementary Fig. S2c). SPINK1 is a trypsin inhibitor and in the normal physiological setting, is almost exclusively expressed in pancreatic cells where SPINK1 is secreted to protect the pancreas from autodigestion. Importantly, SPINK1 is not expressed in the normal liver nor many other major organs and immune cells, making it an attractive therapeutic target (Supplementary Fig. S3a, b). qPCR analysis confirmed our RNA-seq observations and validated the preferential overexpression of *Spink1* in the CD133+ cells and also in the HCC tumor bulk in both HCC tumor models, with expression not altered in the regenerating liver induced by DDC diet or 70% partial hepatectomy (Fig. 2b, c). *Spink1* was also found to be upregulated in a nonalcoholic steatohepatitis (NASH)-HCC mouse model, suggesting that its upregulation is observed across different etiology-driven HCC (Fig. 2d). The expression of *Spink1* and *CD133* correlated positively in both HCC mouse models (Fig. 2e, f). Importantly, genetic depletion of Prom1+ cells (Prom1-DTA) in both HCC mouse models resulted in a significant depletion of *Spink1* as further confirmed by both qPCR and RNAScope analyzes (Fig. 2g–i), implying that CD133 maybe either co-expressed with Spink1 in the same cell and/or CD133 controls the expression of Spink1, though this will have to be further explored. Consistently, *Spink1* was also found to be enhanced in HCC tumors enriched for a CD133+ liver tumor-initiating/propagating cell subset following either 5-FU or cisplatin treatment (Supplementary Fig. S1b).

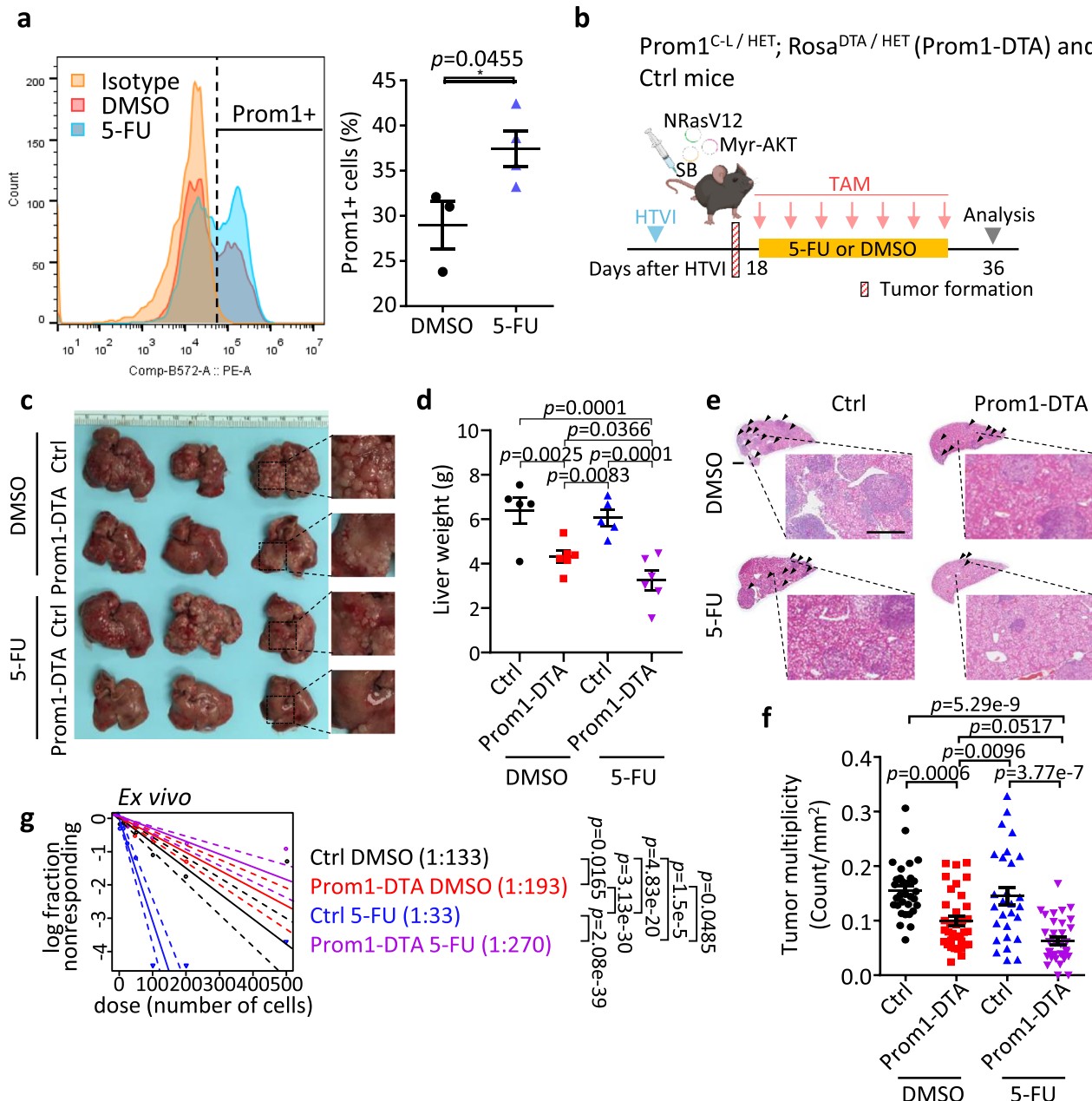

**Fig. 1 | 5-fluorouracil (5-FU) chemotherapy in vivo enrich for a CD133/Prom1+ liver tumor-initiating/propagating cell subset in HCC. a** Flow cytometry analysis for Prom1 expression in mouse NRasV12+Myr-AKT HTVI HCC tumors with DMSO or 5-FU treatment. Isotype-stained cells were used as negative control. **b** Experimental scheme of Prom1+ cells depletion in combination with 5-FU treatment in NRasV12+Myr-AKT HTVI HCC model. Selective depletion of Prom1+ tumor-initiating/propagating cells were achieved by using conditional CreER-induced DTA expression and selective cell death (Prom1-DTA). After tumor initiation, Prom1-DTA and control (Ctrl) mice were treated with 4 mg tamoxifen (TAM), and 5-FU (i.p., 20 mg/kg) or DMSO every other day for two weeks. Liver tissues were collected one week after the last dose of drug treatment. **c** Representative image and **d** quantitative data of liver weights from Ctrl or Prom1-DTA mice treated with DMSO or 5-FU. **e** Representative H&E staining and **f** quantitative data of tumor multiplicity (number of tumor nodules per section) from Ctrl and Prom1-DTA mice after DMSO or 5-FU treatment. Tumors indicated by arrowheads. Scale bar = 1 mm. **g** Ex vivo limiting dilution analysis for frequency of tumor-initiating cells (TICs) in cells harvested from Ctrl and Prom1-DTA mice with DMSO or 5-FU treatments. **a** $n = 3$ mice for DMSO group, 4 mice for 5-FU group; (b-g) $n = 5$ mice for Ctrl DMSO and Ctrl 5-FU groups, 6 mice for Prom1-DTA DMSO and Prom1-DTA 5-FU groups; **f** $n = 31$ views of H&E staining for Ctrl DMSO groups, 35 views of H&E staining for Prom1-DTA DMSO groups, 28 views of H&E staining for Ctrl 5-FU groups, 36 views of H&E staining for Prom1-DTA 5-FU groups; **g** 18–40 replicates in 3 independent experiments. Data were expressed as mean ± s.e.m. Statistical analysis: **a** two-sided Unpaired Student's t-test, **d**, **f** two-way ANOVA, or **g** one-sided Person's χ2 test with 95% confidence intervals. Source data are provided as a Source Data file. An illustration for **b** was created using BioRender.com.

## Overexpression of SPINK1 associates with aggressive clinical features and induces HCC tumor-lineage plasticity

We next explored the involvement of SPINK1 in liver development (Supplementary Fig. S3c). Fetal mouse livers at differential developmental stages were collected[23,24], and the relative expression of *Spink1* was measured by qPCR. *Spink1* expression peaked at embryonic days 16–18 (E16-18) and then rapidly dropped during hepatocyte differentiation (Fig. 3a), which mirrored the expression of several liver progenitor markers that were previously reported, including *Krt19*, *Krt7*, *Sox9* and *Afp*[23,24]. In a separate in vitro differentiation model[25] in which human embryonic stem cells were differentiated into hepatocytes, *SPINK1* was also found elevated in premature hepatocytes (PH),

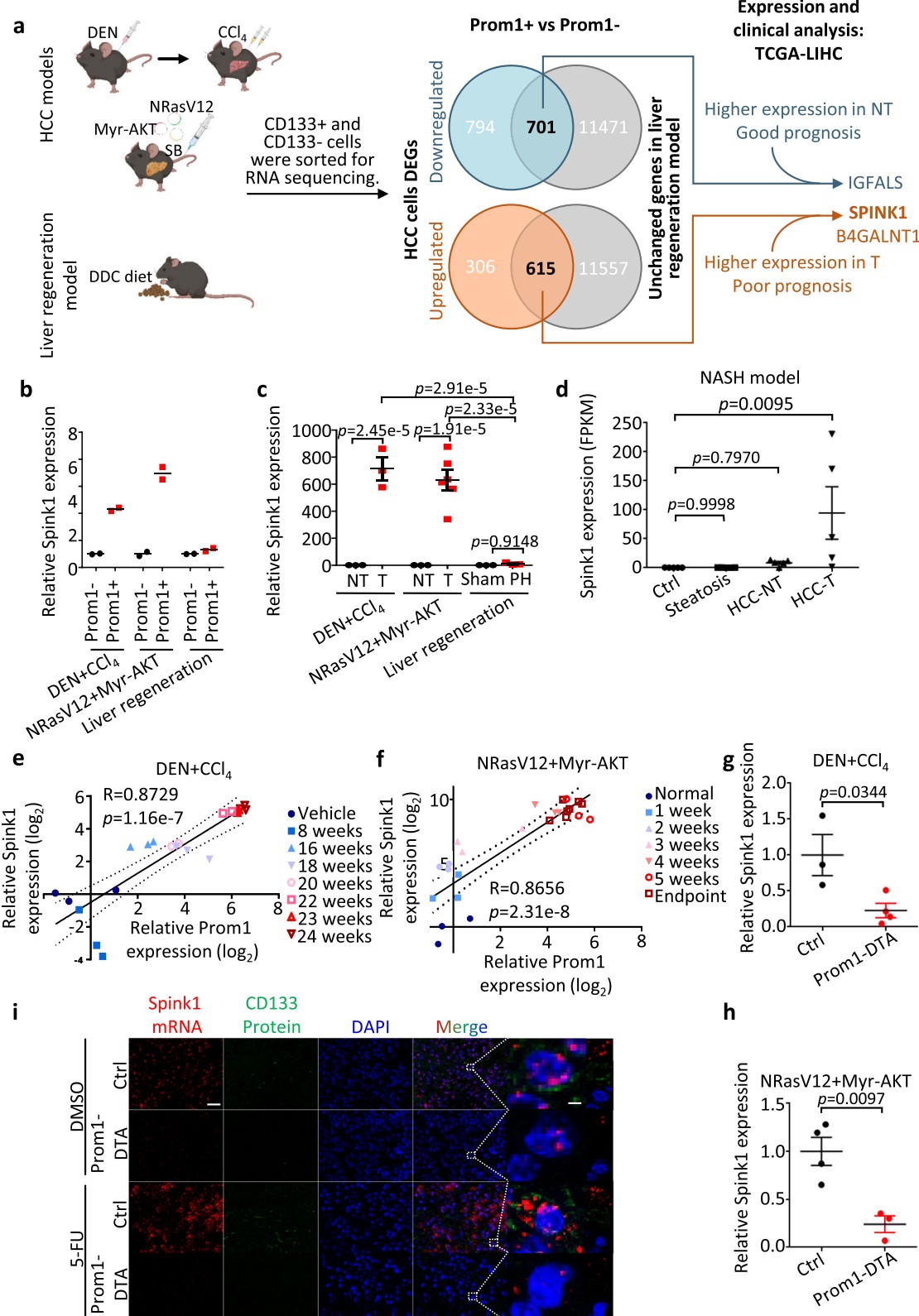

with its expression dropping drastically upon hepatocyte differentiation (Fig. 3b).

To further confirm this in a human setting, we analyzed a recently published scRNA-seq dataset that profiled human liver development[26]. *SPINK1* expression was found elevated in hepatoblast (HB) / fetal hepatocyte (FH) cell clusters and markedly lower in adult hepatocyte (AH) cell clusters, with high *SPINK1* expression correlating well with

hepatic progenitor but not mature hepatocyte markers (Fig. 3c). Using CytoTRACE, a recognized computational framework for predicting differentiation states from analysis of scRNA-seq data[26,27], we found high *SPINK1* expression to overlap with the less differentiated hepatic cells (Fig. 3d). Further analysis of TCGA-LIHC data found *SPINK1* expression to be progressively increased from well to poorly differentiated HCC tumors (histological grade/differentiation stages 1 to 4)

**Fig. 2 | Preferential upregulation of *Spink1* in CD133/Prom1+ tumor-initiating/propagating cells and HCC tumors in mouse models. a** Experimental scheme: differential expressed genes (DEGs) analysis in Prom1+ tumor-initiating/propagating cells using two HCC mouse models (DEN+CCl₄-induced fibrosis-related HCC and NRasV12+Myr-AKT HTVI) and one liver regeneration model (0.1% DDC diet). Transcriptome sequencing identified Prom1+ cells-specific DEGs in HCC tumors, but not in regenerating liver. TCGA-LIHC cohort used for expression and clinical survival analysis to narrow down potential candidate genes. See Supplementary Fig. S2A for detailed data analysis pipelines. SB = Sleeping beauty **b** qPCR validation for *Spink1* expression in the CD133+ and CD133- subpopulations in the three mouse models. **c** qPCR analysis for *Spink1* expression in non-tumor (NT) and tumor tissues (T) of the two HCC mouse models, and in sham or regenerating livers of 70% partial hepatectomy (PH) mouse model. **d** RNA-seq analysis for *Spink1* mRNA expression in various tissues of nonalcoholic steatohepatitis (NASH) mouse model. **e, f** Pearson correlation analysis of *Spink1* and *Prom1* expression in two HCC models, from normal to advanced HCC (●, ■, ▲, ▼, ○, □, △, ▽). **g, h** qPCR analysis for *Spink1* expression in Ctrl and Prom1-DTA mice of the two mouse models. **i** Representative liver tissue images from Ctrl or Prom1-DTA mice of NRasV12+Myr-AKT mouse model treated with DMSO or 5-FU, showing *Spink1* (red) and CD133 (green). DAPI (blue), nucleus. Scale bar in low magnification = 30 μm; high magnification = 2 μm. **b** $n = 2$ mice; **c** $n = 3$ mice (T and NT) of DEN+CCl₄ model and liver regeneration model, $n = 3$ mice (NT), 6 mice (T) of NRasV12+Myr AKT model; **d** $n = 5$ mice; **e, f** $n = 2–7$ mice, **g, h** $n = 3$ mice (Ctrl), 4 mice (Prom1-DTA) of DEN+CCl₄ HCC model, $n = 4$ mice (Ctrl), 3 mice (Prom1-DTA) of NRasV12+Myr AKT HCC model. Data were expressed as mean ± s.e.m. Statistical analysis: **b, c** two-way ANOVA, **d** one-way ANOVA, **e, f** two-sided Pearson correlation analysis, or **g, h** two-sided Unpaired Student's t-test. Source data are provided as a Source Data file. Illustration for **a** was created using BioRender.com.

(Fig. 3e), with high *SPINK1* expression clustering well with hepatic progenitor markers (as opposed to its negative correlation with mature hepatocyte markers) (Fig. 3f). HCC patients with high *SPINK1* expression also enriched for the embryonic stem cell, undifferentiated cancer and HCC stem cell signatures (Fig. 3g). Importantly, enrichment of a stem cell gene signature[28] excluding immune and proliferative genes[29] remained prominent in HCC patients with high *SPINK1* expression, suggesting that correlation of SPINK1 with stemness is likely independent of altered proliferation (Fig. 3g). Further, high *SPINK1* expression was also found in patients that exhibited a high mRNA stemness index as defined by a stronger association with biological processes active in cancer stem cells and with greater tumor dedifferentiation determined by histopathological grade[30] (Fig. 3h). Collectively, these data suggest a link between SPINK1 and oncofetal-stemness characteristics.

We also interrogated publicly available datasets and tissue microarray to compare SPINK1 expression in non-tumor liver and HCC samples. In the TCGA-LIHC dataset, *SPINK1* was found to be frequently overexpressed in HCC compared to its non-tumor liver counterparts, and high *SPINK1* was tightly correlated with worse overall survival (Supplementary Fig. S2b). Immunohistochemistry analysis using a tissue microarray comprising a cohort of 61 paired non-tumor liver and HCC human tissue samples found proteomic level of SPINK1 to also be frequently overexpressed in HCC, with >20% of the cases scored as medium-high expression, while only 1.6% of the non-tumor liver cases scored in the same category (Fig. 3i). Of the 61 paired cases examined, 39 (63.9%) of them exhibited higher SPINK1 expression as compared to its adjacent non-tumor counterpart. High proteomic SPINK1 levels also significantly correlated with a worst overall survival (Fig. 3i). *SPINK1* was also found elevated in nonalcoholic steatohepatitis (NASH)-related human HCC as compared to non-tumor counterparts (Fig. 3j).

### SPINK1 promotes self-renewal and chemoresistance in HCC cells
To test the functional role of SPINK1 in HCC, the relative mRNA expression of *SPINK1* was screened in a series of HCC cell lines (Fig. 4a), where both endogenous and secretory SPINK1 expression followed a similar trend as CD133 expression. SPINK1 was overexpressed and repressed in MHCC97L and Huh7 cells, respectively, by a lentiviral-based approach (Supplementary Fig. S3d). SPINK1 knockdown resulted in a 2-fold decrease in the ability of cells to self-renew, as demonstrated by decreased tumor-initiating frequency in vitro (Fig. 4b). Knockdown of SPINK1 also diminished tumorigenicity in vivo, with a marked decrease in tumor incidence and tumor-initiating frequency (Table 1), concomitant with an increased tumor latency (Table 1, Fig. 4d) and tumor-free survival (Fig. 4e), than HCC cells expressing the non-target scrambled control. Furthermore, knockdown of SPINK1 also significantly reduced stem cell frequency ex vivo when compared to the non-target scrambled control (Fig. 4f). Knockdown of SPINK1 also resulted in a profound decrease in the ability of cells to resist 5-FU and cisplatin, as indicated by enhanced rates of

apoptosis in the knockout clones as compared to the control (Fig. 4g). Consistent functional experimental results were also observed when SPINK1 was overexpressed in MHCC97L cells (Fig. 4c, h).

To evaluate if SPINK1 exerts its functional role also in a secretory manner, we first examined secretory SPINK1 expression in serum HCC clinical samples. Circulating SPINK1 detected in the serum showed a stepwise increase from healthy normal individuals to HCC patients with early and advanced-stage tumors (Fig. 5a). Addition of recombinant SPINK1 in MHCC97L cells promoted tumor-initiating ability (Fig. 5b) and chemoresistance in vitro (Fig. 5d); while conversely, treatment of high-SPINK1 expressing Huh7 cells with a SPINK1 neutralizing antibody[31,32] attenuated such events (Fig. 5c, e).

### EGFR is critical in facilitating SPINK1-mediated hepatocarcinogenesis
Previous studies not in HCC have suggested EGFR to be a potential receptor of SPINK1[31]. Herein, we also explored if SPINK1 would act through EGFR to promote tumor-initiating and chemoresistance properties in HCC. Indeed, co-immunoprecipitation (Co-IP) of SPINK1 and EGFR in HCC cells confirmed their binding (Fig. 6a); while multiplex immunostaining demonstrated their colocalization in human HCC tumor samples (Fig. 6b). To functionally characterize the importance of EGFR in mediating the cancer stemness properties directed by SPINK1, we treated HCC cells with recombinant SPINK1, in the absence or presence of EGFR knockdown (Supplementary Fig. S4a) or FDA-approved EGFR inhibitor, Erlotinib. EGFR knockdown or treatment with Erlotinib abrogated recombinant SPINK1-induced self-renewal and chemoresistance properties (Fig. 6c, e, Supplementary Fig. S5b, d), suggesting SPINK1 regulates HCC through an EGFR axis. Consistent functional experiment results were also observed when SPINK1 was overexpressed in MHCC97L cells with EGFR knockdown or Erlotinib treatment (Fig. 6d, f, Supplementary Fig. S5c, e).

### Transcription factor ELF3 drives SPINK1 expression
In order to investigate the potential upstream regulatory element responsible for SPINK1 overexpression in HCC, we utilized the Gene Transcription Regulation Database (GTRD)[33], a publicly available collection of ChIP-seq data, to predict gene transcription factors binding to the SPINK1 promoter. Our analysis revealed that the cell cycle-related transcription factor ELF3 contains two high-scoring predicted binding sites, including GAAAAGGAAAAAA at -3860 to -3848 and AAGGAAGAAATAA at -2047 to -2035 (Fig. 7a). Like *Spink1*, the preferential overexpression of *Elf3* in the CD133+ cells and also in the HCC tumor bulk in both proto-oncogene driven and inflammation-associated HCC tumor models was also apparent, while *Elf3* expression was not altered or at a much lesser extent in the regenerating liver (Supplementary Fig. S4b, c). Genetic depletion of Prom1+ cells in both HCC mouse models also resulted in a significant depletion of *Elf3*/ELF3 as evidenced by qPCR (Supplementary Fig. S4d) and immunohistochemistry (Supplementary Fig. S4e). Correlation analysis in both

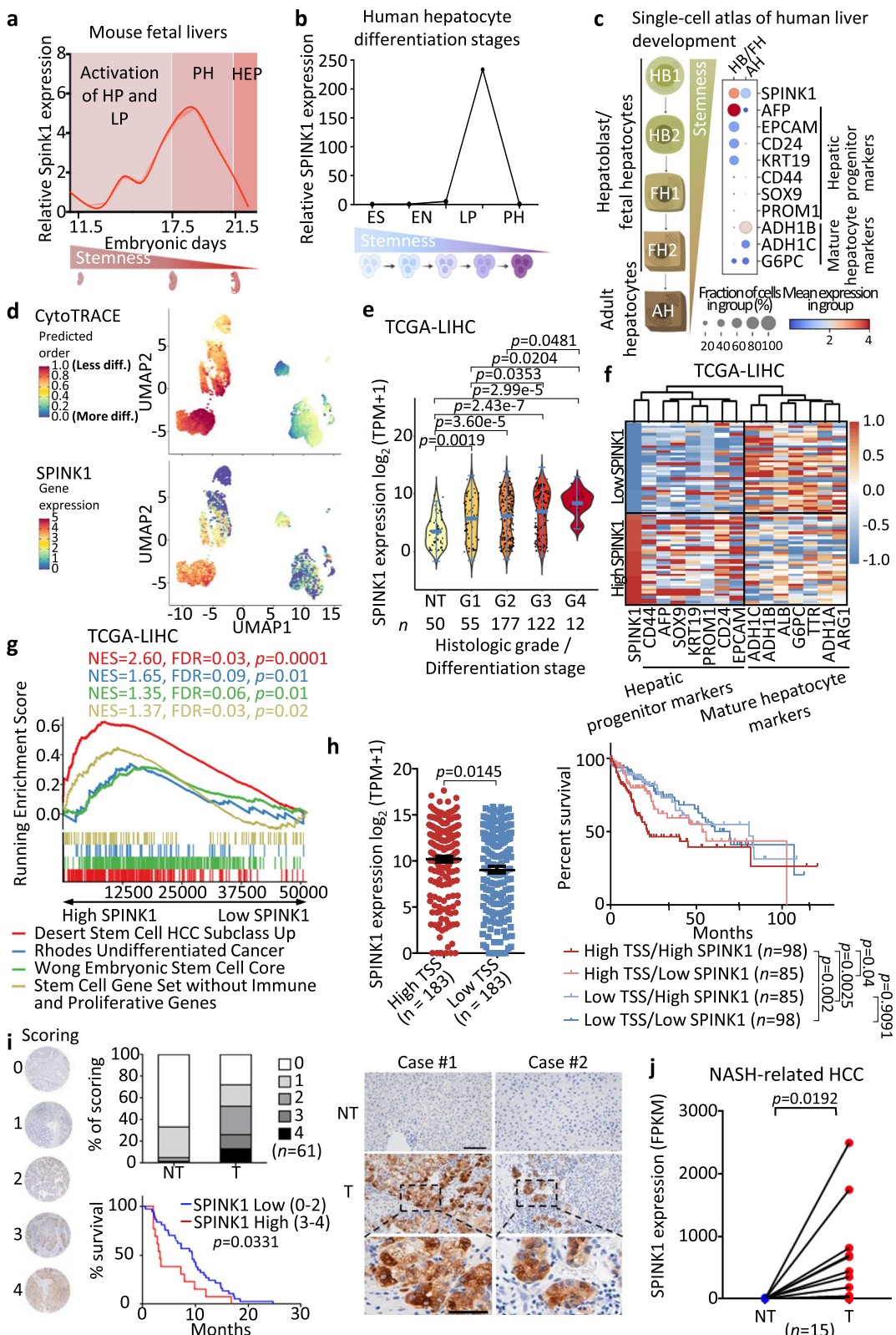

human HCC tumor samples and in the NRasV12+Myr-AKT proto-oncogenes-driven HCC tumor mouse model likewise found ELF3/*Elf3* and SPINK1/*Spink1* expression to positively correlate (Fig. 7b, c). Multiplex staining also confirmed co-localization of SPINK1 and ELF3 in human HCC tumor samples (Fig. 7d). To test the ability of ELF3 to control SPINK1 expression, we generated SPINK1 wild-type and mutants with either one or both predicted binding sites truncated

(Fig. 7e) and then carried out a luciferase reporter assay. A significant reduction in luciferase activity was observed when either one or both of the predicted binding sites were truncated, suggesting that the two predicted ELF3 binding sites are critical in controlling *SPINK1* transcription (Fig. 7f). We also assessed the binding of ELF3 to the SPINK1 promoter by ChIP-qPCR, where the SPINK1 promoter at predicted binding sites S1 and S2 showed a 600- and 1300-fold enrichment

**Fig. 3 | High expression of Spink1/SPINK1 induces HCC tumor-lineage plasticity and is correlated with poor prognosis. a, b** qPCR analysis of *Spink1/SPINK1* expression **a** in fetal mouse livers and **b** from human hepatocyte differentiation model at different developmental stages. ES Embryonic stem cell, EN Endoderm, HP Hepatic progenitor cell, LP Liver progenitor cell, PH Premature hepatocytes, HEP Mature hepatocytes. **c, d** Analysis of a human liver development single-cell atlas[26] showed **c** *SPINK1*, hepatic progenitor and mature hepatocyte markers expression in HB/FH and AH clusters. Dot size = gene expression frequency; color intensity = expression level, and **d** UMAP of (top) CytoTRACE predicted order (more to less differentiated) and (bottom) *SPINK1* expression (low to high), indicated by color from blue to red. **e** *SPINK1* expression in TCGA-LIHC cohort, stratified by histological grading (Edmondson-Steiner 4-tier). NT non-tumor, G1 well differentiated, G2 moderately differentiated, G3 poorly differentiated, G4 undifferentiated. **f–g** Patients in TCGA-LIHC cohort were ranked based on *SPINK1* expression. High *SPINK1* = top 10% *SPINK1*; low *SPINK1* = bottom 10% *SPINK1*. **f** Heatmap showing the clustering of *SPINK1* expression with hepatic progenitor and mature hepatocyte markers. **g** *SPINK1* high tumors showed enrichment of stem cell-related gene sets[28,29] and undifferentiated cancer, via Gene Set Enrichment Analysis (GSEA). NES = Normalized enrichment scores; FDR = False discovery rate. **h** (Left) TCGA-LIHC dataset analysis of *SPINK1* expression with high and low transcriptomic stemness score (TSS). (Right) Kaplan-Meier curve showing the overall survival in different TSS/*SPINK1* groups. **i** (Left) SPINK1 protein expression in paired T and NT HCC clinical biopsies were examined by immunohistochemistry and expression intensities were scored. Kaplan-Meier analysis of the overall survival in patients with high and low SPINK1 scoring in their tumors. (Right) Representative SPINK1 staining images in paired T and NT samples of two HCC patients. Scale bar in low magnification = 100 μm; high magnification = 50 μm. **j** *SPINK1* transcriptomic level in paired T and NT of NASH-related HCCs. **a** 2 replicates; **b** *n* = 1 experiment; **j** *n* = 15 paired T and NT samples. Data were expressed as mean ± s.e.m. Statistical analysis: **g** Kolmogorov–Smirnov test, **h** two-sided Unpaired Student's t-test, **h, i** log-rank test, or **j** two-sided Paired Student's t-test. **e** Data was expressed as a range (min to max), median as the centre, and distribution shape was shown. Statistical analysis: one-way ANOVA. Source data are provided as a Source Data file.

respectively in ELF3 binding compared to ChIP with nonspecific IgG (Fig. 7g). Consistently, stable knockout of ELF3 by lentiviral-based CRISPR-Cas9 knockout strategy resulted in a marked reduction in *SPINK1* transcription and both endogenous and secretory SPINK1 expression levels (Fig. 7h–j). To link this observation to chemoresistance, we also examined the effect of ELF3 knockout on SPINK1 in the presence or absence of 5-FU and cisplatin. Consistent with CD133 expression, both SPINK1 and ELF3 was found upregulated after 5-FU and cisplatin treatment (Supplementary Fig. S4f), with the increase effect abolished upon ELF3 suppression (Fig. 7k−m and Supplementary Fig. S4g).

## SPINK1 promotes self-renewal, dedifferentiation, chemoresistance and tumor initiation through a deregulated ERK-CDK4/6-E2F2 regulatory axis

To decipher the downstream molecular mechanism by which SPINK1 drives HCC, we carried out pathway analysis with HCC samples segregated into high and low *SPINK1*, where enrichment of E2F targets was apparent (Fig. 8a). Given the well-established association between E2F and cell cycle regulation[34], our initial investigation focused on determining if SPINK1 promotes HCC by influencing cell cycle progression. While Western blot found EGFR-downstream effectors including MEK/ERK, G1/S phase-related proteins including CDK4, CDK6, cyclin D1, p-RB and E2F2 (but not E2F1 and E2F3) to be consistently up-regulated upon endogenous SPINK1 overexpression or when HCC cells were treated with recombinant SPINK1, cell proliferation-related proteins including MCM3, PCNA and Cyclin A2, and G1/S phase arrest was not altered (Fig. 8b, c). Similar observations were also noted upon SPINK1 suppression by lentiviral based or neutralizing antibody approaches (Fig. 8b). To further validate the functional roles of this pathway in linking SPINK1, rescue experiments were performed. The utilization of Palbociclib, an FDA-approved CDK4/6 inhibitor, in HCC cells demonstrated its ability to rescue the SPINK1-induced increases in self-renewal (Supplementary Fig. S6a-b), chemoresistance (Supplementary Fig. S6c-d), and activation of p-RB-E2F2 signaling (Supplementary Fig. S6e). Furthermore, GSEA analysis of HCC patients in the TCGA-LIHC cohort revealed no significant enrichment of gene signatures associated with G1/S transition or cell proliferation in HCC cells harboring high *SPINK1* expression (Fig. 8d). Of note, similar deregulation in ERK-CDK4/6-E2F2 signaling axis could also be validated in HCC cells treated with either recombinant SPINK1 or have SPINK1 stably overexpressed, with concomitant EGFR repressed, demonstrating EGFR as an important upstream regulator of the pathway (Supplementary Fig. S5a, f).

Of interest, studies have now shown a link between cell cycle-related proteins, self-renewal and oncogenic dedifferentiation. For instance, RB inactivation is found to promote the reprogramming of differentiated cells to a pluripotency state[35]; while cyclin D is found to be essential for pluripotency maintenance in human embryonic stem cells by preventing endoderm differentiation[36]. Inhibition of CDK4/6 and cyclin D has also been shown to result in differentiation of endoderm cells to hepatic cells[36]. Analysis of TCGA-LIHC data also found high *SPINK1* expression to cluster well with E2F2 target genes related to stemness, dedifferentiation and chemoresistance (Fig. 8f), but not E2F2 target genes related to cell cycle regulation (Supplementary Fig. S6f). Consistent results were observed in the alteration of *E2F2* and its target genes, including *FGFR3*, *SPHK1*, and *MYBL2*, in HCC cells upon manipulating the expression of SPINK1 (Fig. 8g–j). Analysis of TCGA-LIHC data found HCC patients that displayed high expression of the *ELF3 + SPINK1 + E2F2* signature exhibited a worst prognosis than those HCC patients with HCC patients with a low expression of the same three gene signature, drawing clinical relevance to further support our functional observations (Fig. 8e and Supplementary Fig. S6g). Further analysis of human HCC data also found high *SPINK1* expression to correlate with gene signatures relating to tumor recurrence, hepatic progenitor, poorly differentiated HCC and chemoresistance, but not cell cycle, G1 phase or G1/S transition (Supplementary Fig. S7).

## Therapeutic targeting of SPINK1 may represent a promising treatment option for chemoresistant HCC

We next interrogated SPINK1 dependency in a proof-of-principle therapeutic experiment. Frequent activation of both the RAS/MAPK and PI3K/AKT pathways is documented in almost 50% of HCC patients[21]; thus, we chose to perform our proof-of-principle therapeutic targeting of the SPINK1 model in the NRasV12+Myr-AKT HTVI-induced HCC model. We showed that SPINK1 was elevated in a stepwise manner in NRasV12+Myr-AKT HTVI-driven HCC (Fig. 2f). We treated HCC tumors with either chemotherapy alone or in combination with lentiviruses encoding Spink1 knockdown with the latter delivered intravenously into the mice (Fig. 9a). Lentivirus encoding Spink1 knockdown and chemotherapy combination produced a maximal suppression effect, as evidenced by reduced tumor size (Fig. 9b), suppressed liver weight (Fig. 9c) and prolonged survival time (Fig. 9d). The combination treatment is also effective in decreasing tumor-initiating cell frequency as measured by subjecting the harvested tumor cells for limiting dilution spheroid formation assay ex vivo (Fig. 9e). Of interest, while the relative expression of stemness markers was significantly increased upon 5-FU treatment alone, combination treatment resulted in a significant decrease in stemness and increase in differentiation gene expression (Fig. 9f). RNAScope was performed to show the successful knockdown of Spink1 in vivo, while chemotherapy treatment alone would lead to an increase in *Spink1* expression in control group but not in the Spink1 suppressed group (Fig. 9g).

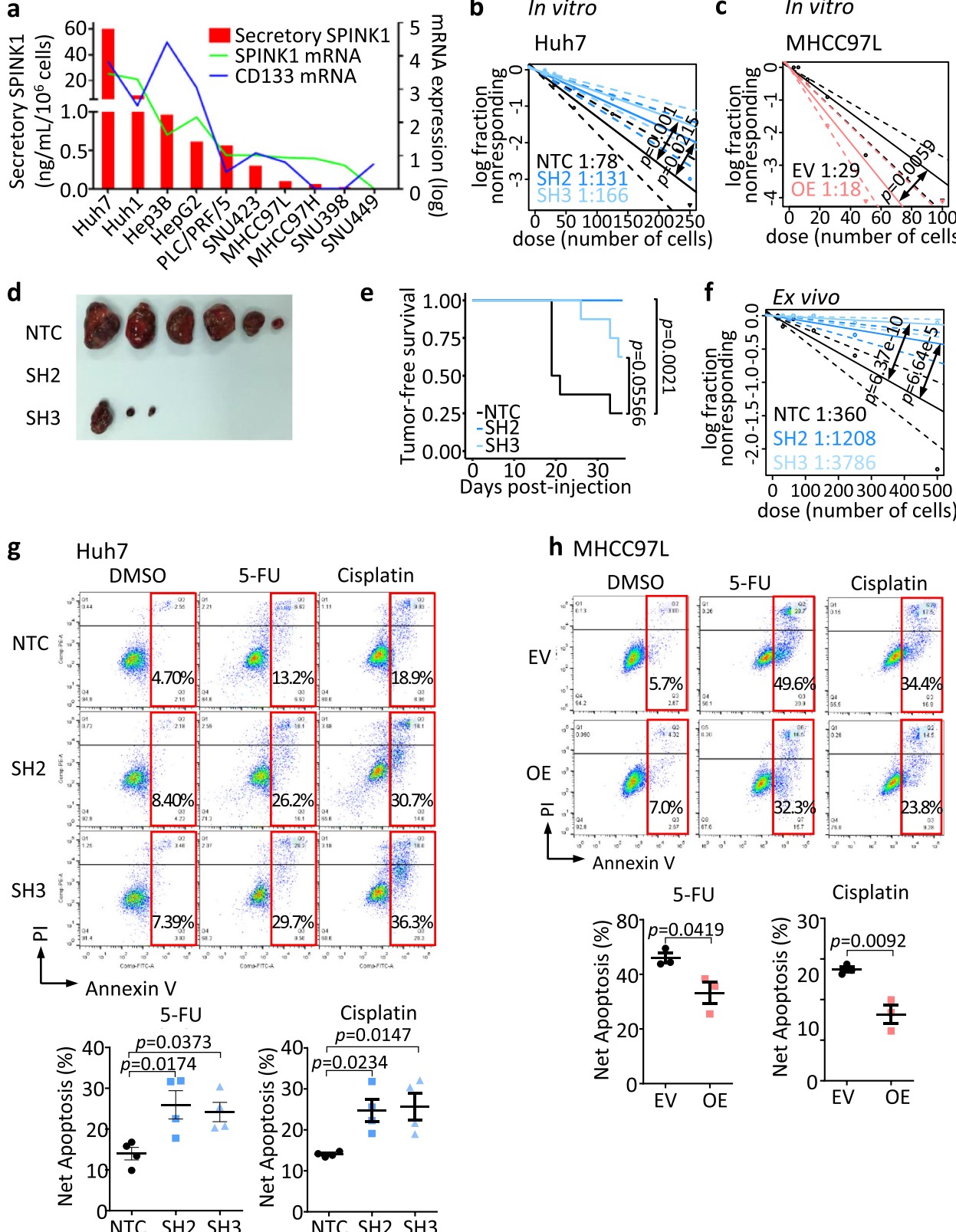

## Discussion

Tumor lineage plasticity has become increasingly recognized as a prominent mechanism contributing to therapeutic resistance[37,38]. There is a striking resemblance between tumor development and developmental processes, potentially influencing the fate of tumor cells and their capacity for lineage plasticity. One feature shared by

tumor-lineage plasticity and developmental processes is the activation of potential tumor-initiating/propagating cells in the tumor and re-expression of stem/progenitor cell markers, which usually remain low, if not absent, in normal terminally differentiated cells[39]. A number of intrinsic and extrinsic regulators have now been demonstrated to play important roles in driving HCC tumor lineage plasticity, including

**Fig. 4 | SPINK1 promotes self-renewal and chemoresistance in HCC cells.**
**a** SPINK1 and CD133 expression in a human HCC cell line panel. Red bars indicate secretory SPINK1 measured by ELISA. Lines indicate log transformed mRNA expression levels. **b, c** In vitro limiting dilution assays for evaluating self-renewal abilities in MHCC97L and Huh7 cells with or without SPINK1 expression manipulated. **d, e** In vivo limiting dilution assays, in serial passage, for evaluating tumor-initiating cells (TICs) frequency of Huh7 cells with either NTC, SH2, or SH3. **d** Tumor incidence, average latency, and estimated tumor-initiating frequency data. **d** Representative image of tumors formed by 50,000 Huh7 cells transplanted in NOD-SCID mice. **e** Kaplan–Meier curves showing the percentage of tumor-free survival of NOD-SCID mice transplanted with 50,000 Huh7 cells. **f** Ex vivo limiting dilution assay of tumors harvested from the primary implantation. **g, h** Cell

apoptosis upon 5-FU or cisplatin treatment as demonstrated by Annexin V-PI flow cytometry analysis, in **g** Huh7 cells or **h** MHCC97L cells with or without SPINK1 gene manipulated. NTC = Non-target scrambled control; SH2 and SH3 = shRNA clones targeted for SPINK1 knockdown; EV = Empty vector control; OE = SPINK1 over-expression. **a** 2 replicates, $n = 1$; **b, c** 20 replicates in 3 independent experiments; **d, e** $n = 4$ mice for primary implantation, 7 mice for secondary implantation; **f** 20 replicates; **g** $n = 4$ independent experiments; **h** $n = 3$ independent experiments. Data were expressed as mean ± s.e.m. Statistical analysis: **b, c, f** one-sided Person's $\chi^2$ test with 95% confidence intervals, **e** log-rank test, or **g** one-way ANOVA, or **h** two-sided Unpaired Student's t-test. Source data are provided as a Source Data file.

## Table 1 | In vivo limiting dilution assays of SPINK1 knockdown

| Huh7 | Tumor incidence | | | Average latency (Days) | | | Estimated TIC frequency |
|---|---|---|---|---|---|---|---|
| **Primary implantation** | | | | | | | |
| Cell numbers | **5k** | **10k** | **50k** | **5k** | **10k** | **50k** | |
| NTC | 5/8 | 4/8 | 6/8 | 31.9 | 32.6 | 25 | 1/18262 |
| SH2 | 1/8 | 1/8 | 0/8 | 34.1 | 34.8 | - | 1/256230 ($p = 8.36e-6$) |
| SH3 | 1/8 | 3/8 | 3/8 | 34.1 | 33.3 | 33.6 | 1/59524 ($p = 0.0444$) |
| **Secondary implantation** | | | | | | | |
| Cell numbers | **5k** | **10k** | | **5k** | **10k** | | |
| NTC | 10/14 | 13/14 | | 20.3 | 19.1 | | 1/3897 |
| SH2 | 3/14 | 6/14 | | 24.4 | 22 | | 1/18836 ($p = 5.63e-5$) |
| SH3 | 10/14 | 7/14 | | 19.8 | 22.5 | | 1/8271 ($p = 0.0437$) |

In vivo limiting dilution assays, in serial passage, for evaluating tumor-initiating cells (TICs) frequency of Huh7 cells with either non-target scrambled control (NTC) or SPINK1 knockdown lentiviral transduction (SH2 and SH3). Tumor incidence, average latency and estimated tumor-initiating frequency data. $n = 4$ mice for primary implantation; $n = 7$ mice for secondary implantation. Significance was calculated by one-sided Pearson's $\chi^2$ test comparison to NTC.

PGC7[40], CLDN6[41], and the interplay of TGF-β and GDF-1[24], etc. Here, we report on the role of endogenous/secretory SPINK1 in inducing HCC to a more stem/progenitor state and making HCC cells resistant to chemotherapy. To date, there has been controversy regarding the identification of CSCs primarily due to the technical differences in experimentation and CSC assays, etc.[42]. How tumor-initiating cells are defined or supported in this study is by various means, including correlation with CD133 which is a widely reported surrogate marker of liver cancer stem cells[5,6,8], (ii) correlation with stemness/undifferentiated signatures defined by CytoTRACE[27] and cancer stemness index[30], etc. Most importantly, (iii) tumor-initiating ability (self-renewal) is functionally defined by in vitro and in vivo limiting dilution assays, experiments that are now widely accepted in the field[43].

As evidenced by scRNA-seq profiles of normal tissues, *SPINK1* is most prominently expressed in the pancreas, but nearly undetectable in the colon, liver, rectum, small intestine and stomach[44], making it an attractive target for cancer therapeutic targeting. SPINK1 has been reported to modulate reproduction, differentiation and repair of normal tissues[45]. SPINK1 has also been widely associated with a range of malignant tumors, including HCC, where it has been reported to be more highly expressed in HBV and HCV-related HCC[10–12], and that SPINK1 serum levels alone or in combination with AFP, is more accurate for early HCC diagnosis than as compared to just AFP alone[13]. Functionally, SPINK1 has also been found to accelerate cell proliferation and increase metastatic and invasive potential[14–16]. However, there have been limited studies on the role of SPINK1 in the context of tumor plasticity and cancer stemness. CD133/PROM1 is one of the best-studied functional and phenotypic markers of liver tumor-initiating/propagating cell subset[20]. By lineage tracing approach, we recently showed Prom1 in HCC tumors to mark proliferative tumor-propagating cells with cancer stem cell-like properties. Labeled Prom1+ cells exhibit increasing tumorigenicity in 3D culture and allotransplantation, as well as potential to form cancers of differential lineages on transplantation. Depletion of Prom1+ cells impedes tumor growth and reduces malignant cancer

hallmarks in both HCC models[8]. Most interestingly, we found Prom1+ HCC cells to follow a dedifferentiation trajectory and that Prom1-lineage gene signature predicts poor prognosis in HCC[8]. In this study, we find chemotherapy treatment to induce a CD133 and SPINK1 population in HCC and that Prom1-lineage ablation in HCC tumors results in a marked suppression of SPINK1 and can sensitize HCC cells to chemotherapy. SPINK1 is strongly associated with aggressive cancer features and poor tumor differentiation in HCC as well as a more stem/progenitor and less differentiated state in liver development. To ensure the association between SPINK1 and stemness is not a result of misinterpretation of its association with proliferation, we exclude cell proliferate markers from the originally measured gene sets and reassess the contribution of proliferation-related genes to the observed stem cell signature enrichment. Despite having the proliferation markers omitted, HCC tumors with high SPINK1 still show highly significant enrichments of stem cell signatures, suggesting that the stemness signature is independent of the proliferation status. As we previously found CD133/Prom1 to mark proliferative tumor-propagating cells, one would expect that more proliferative cells would be more easily subjected to chemotherapy killing. Yet, we show here that 5-FU or cisplatin chemotherapy enrich for CD133/Prom1, suggesting that while CD133 may be proliferative, they may also harbor other mechanisms (e.g., SPINK1-mediated tumor plasticity) that enable them to resist standard chemotherapy.

Manipulation of endogenous/intracellular and secretory/circulating levels of SPINK1 in HCC identified its functional role in driving self-renewal and chemotherapy resistance. We also show in a proof-of-concept model that endogenous depletion of hepatic SPINK1 in an immunocompetent HCC mouse model can impede tumor growth, and sensitize the tumor to chemotherapy. Of interest, depletion of hepatic SPINK1 does not diminish the fraction of CD133 liver tumor-initiating/propagating cells though it does reverse the HCC cells to a less stem and more differentiated state as well as decrease the tumor-initiating potential of the HCC cells. Noteworthily, we also test the ability of SPINK1 manipulation to alter HCC cells' sensitivity to tyrosine kinase

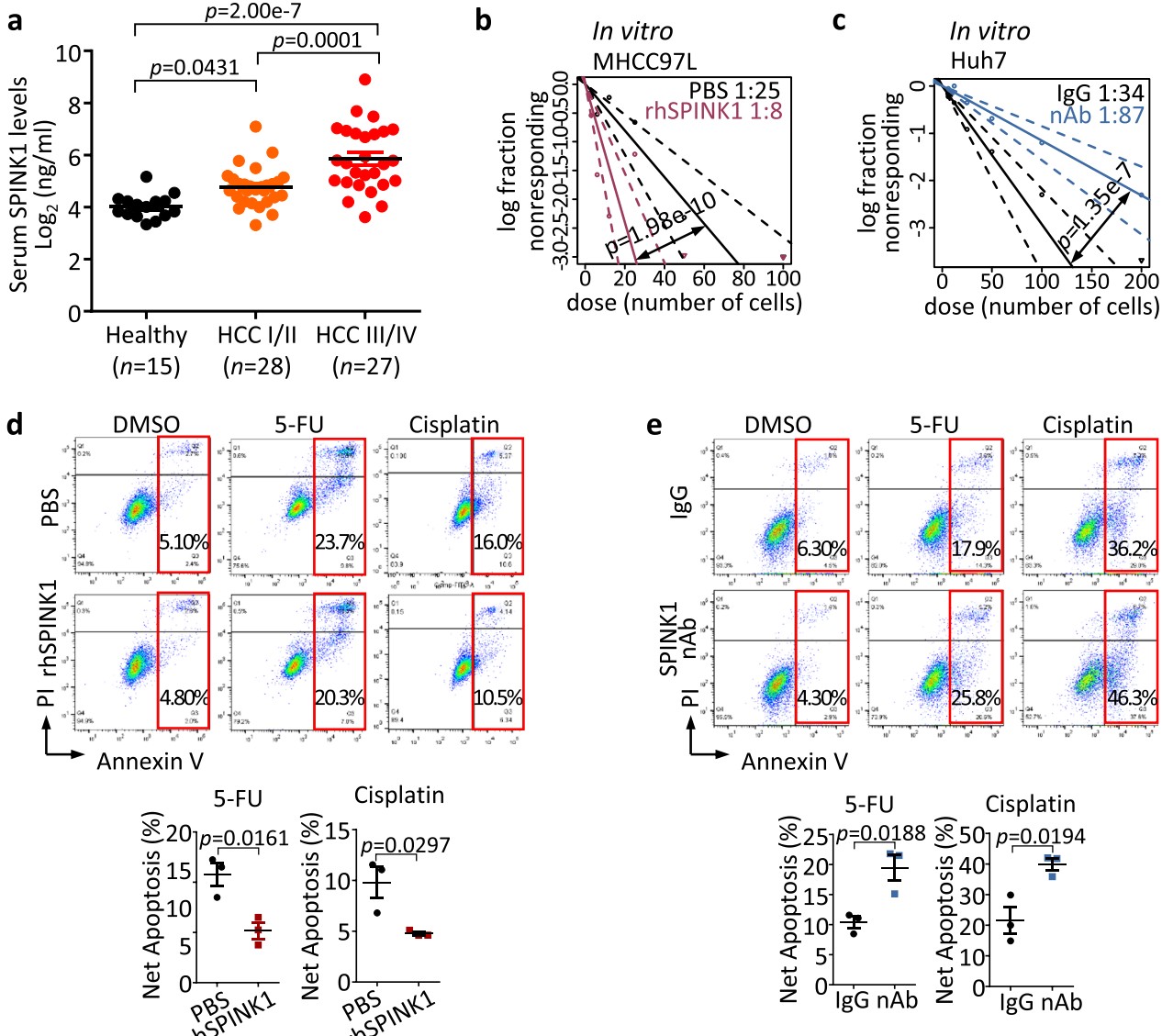

**Fig. 5 | Secretory SPINK1 promotes self-renewal and chemoresistance in HCC cells. a** Secretary SPINK1 was detected in the serum of HCC patients ($n = 55$) compared with that of healthy donors ($n = 15$) by ELISA. HCC I/II = early- to mid-stage HCC, HCC III/IV = late-stage HCC by UICC 7th Edition Staging System for HCC. **b, c** In vitro limiting dilution analysis for frequency of TICs of **b** MHCC97L cells treated with recombinant SPINK1 (rhSPINK1) and **c** Huh7 cells treated with SPINK1 neutralizing antibody (nAb) compared to PBS and IgG, respectively. **d, e** Cell apoptosis upon 5-FU or cisplatin treatment as demonstrated by Annexin V-PI flow cytometry analysis, in **d** MHCC97L cells treated with rhSPINK1 and **e** Huh7 cells treated with nAb. PBS = PBS control; IgG = Isotype control antibody. **b, c** 30 replicates in 3 independent experiments; **d, e** $n = 3$ independent experiments. Data were expressed as mean ± s.e.m. Statistical analysis: **a** one-way ANOVA, **b, c** one-sided Person's χ2 test with 95% confidence intervals, or **d, e** two-sided Unpaired Student's t-test. Source data are provided as a Source Data file.

inhibitor sorafenib, but our data does not show it to be effective. Previous studies in prostate and breast cancers have also demonstrated the efficacy of a monoclonal SPINK1 neutralizing antibody to decrease tumor growth as well as metastatic abilities[31,46]. As a next step to bridge our findings to a more pre-clinical setting, it will be worthwhile to utilize this SPINK1 neutralizing antibody and test its effect in HCC patient-derived xenografts, alone or in combination with chemotherapy. Notably, chemotherapy treatment alone enriches CD133+ liver cancer stem cells and increases SPINK1 expression, suggesting why some HCC patients exhibit resistance to chemotherapy. Therefore, HCC patients with transarterial chemoembolization (TACE) resistance and high serum SPINK1 levels could potentially benefit from monoclonal SPINK1 neutralizing antibodies to sensitize cells to TACE and improve the overall treatment response. It will be interesting to also further explore how the suppression of SPINK1 affects the tumor microenvironment (TME), including immune cells; as well as what

other cell types in the niche may secrete SPINK1 to promote HCC. In this connection, Lu et al. previously found inflammatory SPINK1 to prevent cytolytic granule granzyme A-mediated apoptosis/immune-killing[47]; and that SPINK1 secreted by stromal cells in a damaged tumor microenvironment following chemotherapy, can promote more aggressive cancer phenotypes[32]; additionally, Jia et al. demonstrated SPINK1 as a potential biomarker for the early detection, targeted therapy, and prediction of immune checkpoint blockade (ICB) treatment response in HCC patients[48]. By analyzing a scRNA-seq of a publicly available dataset (GSE151530)[49], *SPINK1* is found to be primarily expressed in malignant cells, with minimal expression in immune cells and stromal cells. These findings suggest that SPINK1 is not only expressed in liver tumor cells but also in other cell types within the TME (Supplementary Fig. S8a-c). Moreover, RNAScope with multiplex IHC staining demonstrates that while immune cells and stromal cells do express SPINK1/Spink1 (Supplementary Fig. S8d-f), only fibroblast

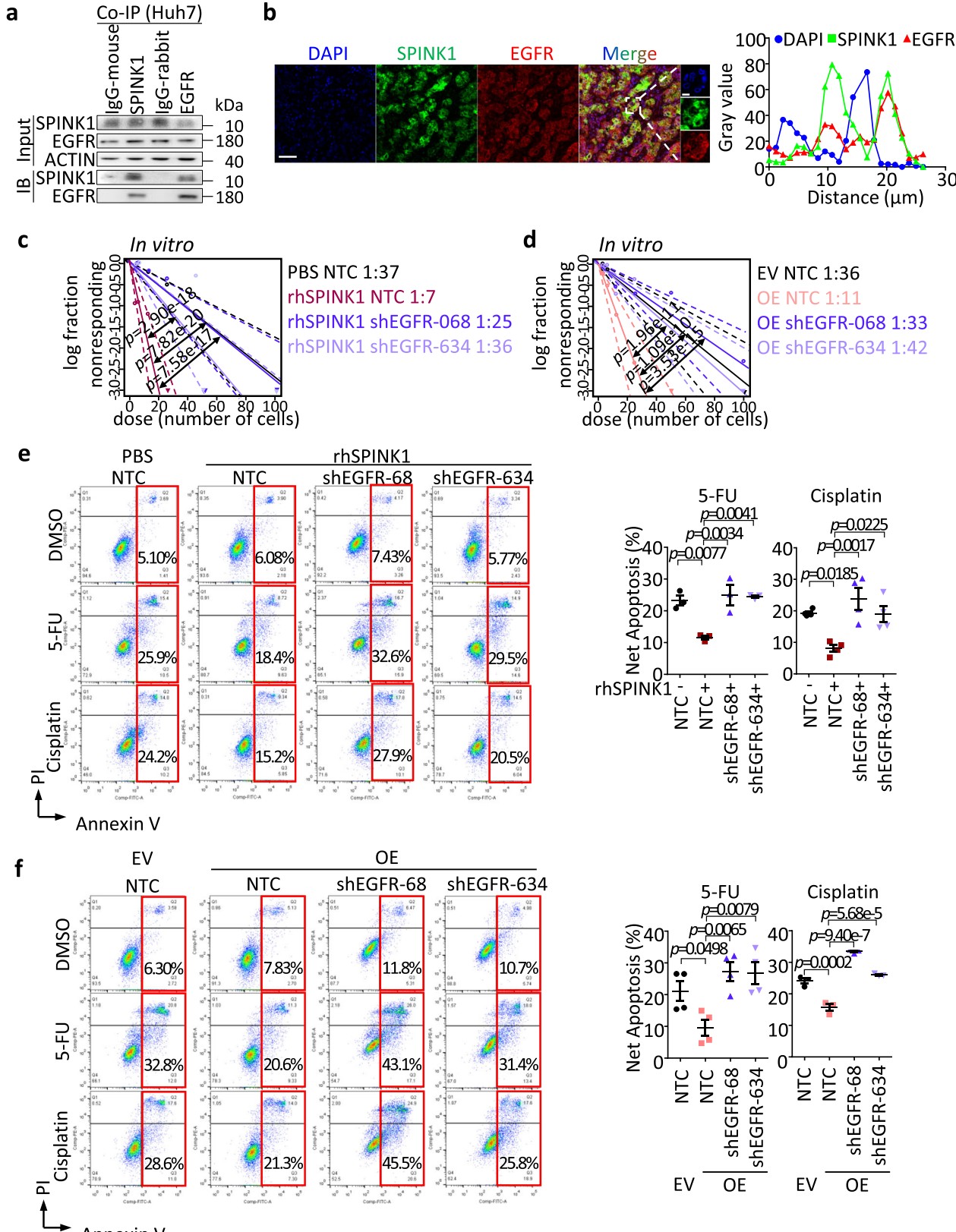

cells show an increase in *Spink1* expression following chemotherapy treatment in HCC (Supplementary Fig. S8f). This does suggest further studies of the functional role of fibroblast-secreted SPINK1 in HCC and the combination therapy with immune checkpoint blockade and/or chemotherapy is indeed warranted. Further, since we have identified EGFR, to be at least one of the binding partners to which SPINK1 signals in HCC, it may be worthwhile to explore on the application of EGFR inhibitor to possibly widen the therapeutic window for chemotherapy in the clinic. To date, none of the EGFR inhibitors (e.g. cetuximab, gefitinib, afatinib, dacomitinib, osimertinib and neratinib) have been approved for treatment of HCC. Yet it will be interesting to examine if EGFR inhibition can induce tumor-lineage plasticity in HCC.

**Fig. 6 | EGFR is critical for SPINK1 to drive HCC. a** Coimmunoprecipitation (Co-IP) analysis for validation of EGFR as an interacting protein partner of SPINK1 in Huh7 cells. IgG-mouse = Mouse isotype control antibody; IgG-rabbit = Rabbit isotype control antibody; IB = Immunoblotting. **b** (Left) Co-staining of SPINK1 (green) and EGFR (red) protein in human HCC tumor by multiplex immunohistochemistry (IHC). DAPI (blue), nucleus. Scale bar: 50μm. (Right) Histogram representation of line scan analysis for quantification of SPINK1 (green, ■), EGFR (red, ▲) and DAPI (blue, ●). **c, d** In vitro limiting dilution analysis for frequency of TICs of **c** MHCC97L with rhSPINK1 treatment or **d** SPINK1 overexpression (OE) in the presence or absence of shRNA against EGFR stably transduced (shEGFR-068 and shEGFR-634).

**e, f** Cell apoptosis upon 5-FU or cisplatin treatment as demonstrated by Annexin V-PI flow cytometry analysis, in **e** MHCC97L cells treated with rhSPINK1 or **f** with SPINK1 overexpression in the presence or absence of EGFR knockdown. **a** $n = 3$ independent experiment; **b** $n = 2$ human HCC tumor samples; **c, d** 30 replicates in 3 independent experiments; **e** $n = 3$ independent experiments for 5-FU group, 4 independent experiments for cisplatin group; **f** $n = 4$ independent experiments for 5-FU group, 3 independent experiments for cisplatin groups. Data were expressed as mean ± s.e.m. Statistical analysis: **c, d** one-sided Person's χ2 test with 95% confidence intervals, or **e, f** one-way ANOVA. Source data are provided as a Source Data file.

Our current study identifies the involvement of the cell cycle regulatory pathway (ERK-CDK4/6-E2F2) in SPINK1-mediated tumor plasticity in HCC. Cell cycle proliferation is an obvious phenotype to measure, yet we are not able to show a significant alteration of G1/S arrest. Upon further reading, a multitude of research studies have shown that proteins involved in the cell cycle, for instance CDK1, cyclin Ds and cyclin Es, are essential for sustaining stem cell pluripotency by stimulating genes that control pluripotency, like OCT4, SOX2 and NANOG[50-52]. Moreover, control of cell cycle is also fundamental for determining cell fate. Pauklin et al. indicated that cyclin Ds were able to hinder the differentiation of embryonic stem cells into hepatocytes by activating CDK4/6 and obstructing endoderm differentiation[50]. Further, activation of cell survival pathways, including those related to cell cycle, has also been shown to be linked to chemoresistance. For instance, Luo et al. showed that upon ARID1A depletion in squamous cell carcinoma, it would trigger the phosphorylation of RB by CDKs, which would then result in the release of E2F1 to activate c-myc expression, leading to an increase in stemness related proteins such as NANOG and SOX2, thus resulting in chemotherapy resistance[53].

The functional enrichment of CD133/Prom1 following 5-FU or cisplatin treatment, as observed in previous and our present studies, suggests that many different mechanisms are likely contributing to its enrichment. In addition to SPINK1-mediate tumor plasticity, other signaling pathways such as the NOTCH pathway have been reported to be activated in CD133 + HCC cells after 5-FU treatment[54]. In our previous investigation, we observed the preferential activation of Akt/PKB and upregulation of Bcl-2 in CD133 + HCC cells contributing to their resistance to 5-FU treatment[6]. Moreover, the TME is known to influence CD133 expression and enrichment. Our prior research demonstrated that chemotherapy treatment enriched THBS2-deficient CD133+ liver CSCs and promoted HCC progression through matrix softness-induced histone H3 modifications[7]. Further studies exploring the interplay between CD133 + HCC cells and the TME will elucidate the impact of these factors on CD133 expression and its enrichment following chemotherapy. Although our present study focuses on SPINK1-mediated tumor plasticity as one potential mechanism contributing to the functional enrichment of CD133+ cells following chemotherapy, other mechanisms, including survival signaling pathways and the TME, may also play significant roles. Herein, we show SPINK1 overexpression in the CD133+ liver tumor-initiating/propagating subset of HCC tumors to resist 5-FU and cisplatin treatment through activating ERK-CDK4/6-cyclinD1-E2F2 regulatory mechanism that eventually leads to maintenance of a more stemness and dedifferentiated state. Lentivirus-mediated suppression of SPINK1 reduces stemness and induces a more differentiated HCC tumor; while combination therapy with 5-FU leads to a more maximal suppression, driving the HCC cells toward a more differentiated lineage (Fig. 9h).

Our study presents several noteworthy findings in the field of HCC. It reveals a significant association between SPINK1 expression and the less differentiated HCC tumors, suggesting its potential as a biomarker for tumor aggressiveness. We also elucidate a unique mechanism by which SPINK1 promotes chemoresistance in HCC through the induction of tumor plasticity, providing insights into the underlying processes of therapeutic response. These findings advances our understanding of the modulation of therapeutic outcomes in HCC by SPINK1.

## Methods
### Study approval
Formalin-fixed paraffin-embedded primary human HCC and adjacent non-tumor liver tissue samples and serum samples of HCC patients were obtained from Queen Mary Hospital with written informed consent obtained from all patients and protocol approved by the Institutional Review Board of the University of Hong Kong/ Hospital Authority Hong Kong West Cluster. Tissue microarray (TMA) was obtained from Professor Jing-Ping Yun at the Sun Yat-sen University Cancer Centre in Guangzhou, China, with the approval of the Institutional Review Board for ethical review from the Sun Yat-sen University in Guangzhou, China with written informed consent from all patients. License to conduct experiments on animals was obtained from the Department of Health, Hong Kong SAR. All animal study protocols were approved by and performed in accordance with the Committee of the Use of Live Animals in Teaching and Research (CULTAR) at the University of Hong Kong and the Animals (Control of Experiments) Ordinance of Hong Kong.

### Cell lines
293 T cells (CRL-3216), 293 T/17 cells (CRL-11268) and HCC cell lines Hep3B (HB-8064), HepG2 (HB-8065), SNU423 (CRL-2238), SNU398 (CRL-2233), SNU449 (CRL-2234) and PLC/PRF/5 (CRL-8024) were purchased from American Type Culture Collection (ATCC). HCC cell lines Huh1 (JCRB0199) and Huh7 (JCRB0403) were purchased from the Japanese Collection of Research Bioresources (JCRB) Cell Bank. 293FT (R70007) cells were purchased from Invitrogen. HCC cell lines MHCC97H and MHCC97L were obtained from the Liver Cancer Institute, Fudan University. The cell lines used in this study were authenticated by STR profiling and tested for the absence of mycoplasma contamination.

### Transcriptome sequencing
Tissues from mice were snap-frozen. Total RNA was extracted using the RNeasy Kit (Qiagen, 74134). Samples with an RNA integrity number (RIN) value over 8 were used. Poly-A mRNA library was prepared with 200 ng total RNA by KAPA Standard mRNA-Seq Kit (KAPA Biosystems, KR0960-v3.15). Libraries were then subjected to paired-end 101 bp sequencing using HiSeq SBS Kit v4 (Illumina, FC-401) and a cluster was generated by HiSeq PE Cluster Kit v4 cBot (Illumina, PE-401-4001). Each sample had an average throughput of 12.8GB and a total throughput of 17.8 in sequence quality. An average of 94% of the bases achieved a quality score of Q30, which denoted the accuracy of a base call to be 99.9%. Sequencing reads were filtered for adapter and low-quality sequences and then aligned to Mouse Genome GRCm38 by STAR (version 2.5.2). Gene counts were calculated by RSEM (version 1.2.31) and raw counts were normalized. Differentially expressed genes (DEGs), which were defined as false discover rate (FDR) < 0.05, were identified by EBSeq2 (version 1.10.0).

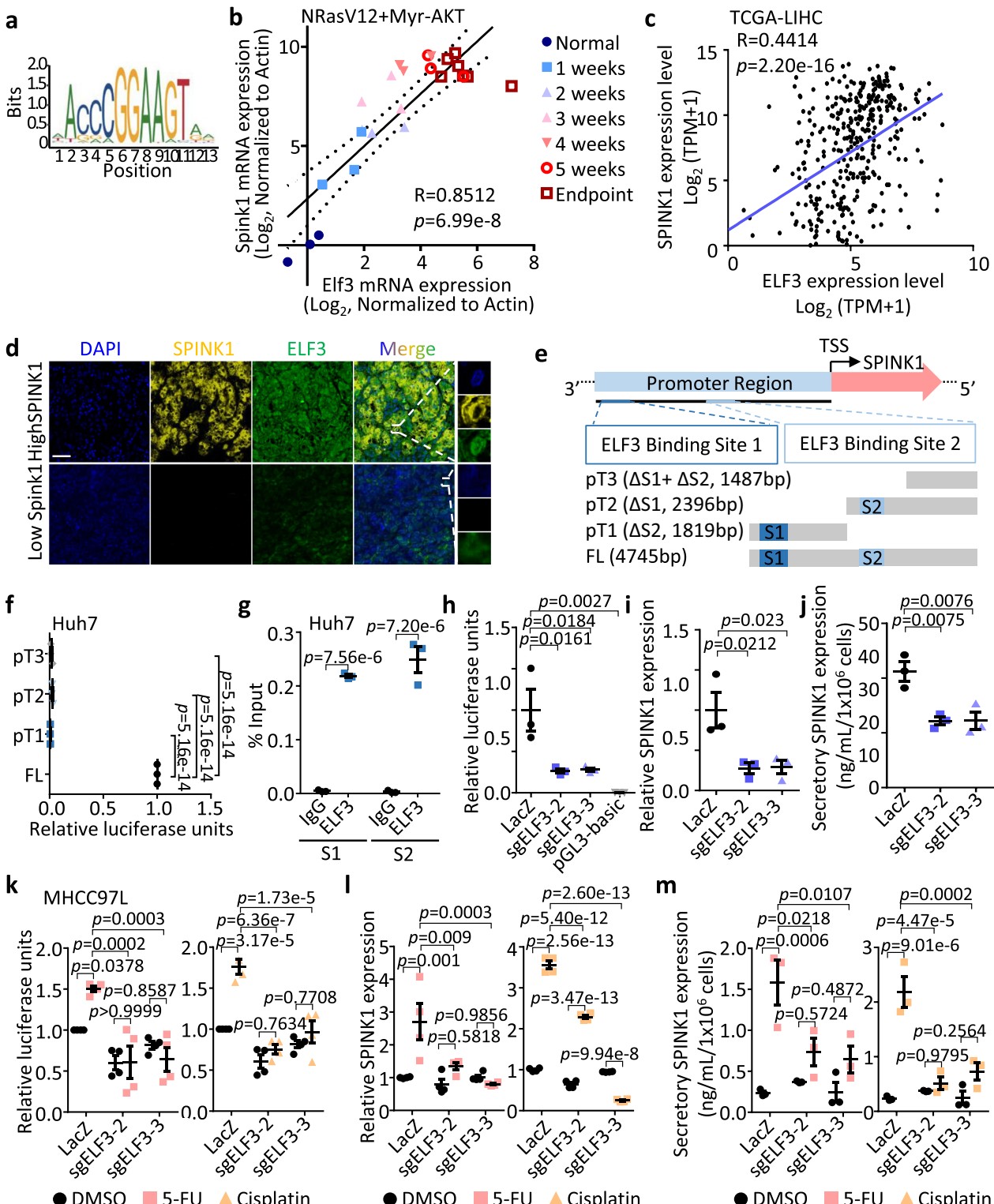

TCGA (The cancer genome atlas)
Gene expression profiling studies were analyzed for the expression of *SPINK1*, *ELF3*, hepatic progenitor markers, mature hepatocyte markers, and E2F target genes transcripts available under Liver Hepatocellular Carcinoma (LIHC) of the TCGA Research Network. For histological grading (Edmondson-Steiner 4-tier), non-tumor liver and HCC tissue samples were analyzed for the expression of *SPINK1* transcripts available under the TCGA-LIHC cohort. For Gene Set

Enrichment Analysis (GSEA) and heatmaps showing the expression of hepatic progenitor markers, mature hepatocyte markers, and E2F target genes, high and low SPINK1 expression was defined as the top 10% and bottom 10% of expression in HCC patients in the TCGA-LIHC. For survival analysis, high and low SPINK1 expression and transcriptomic stemness score (TSS) were defined as the top 25% and bottom 25% of expression in HCC patients in the TCGA-LIHC cohort.

**Fig. 7 | Transcription factor ELF3 drives SPINK1 expression. a** Consensus binding motif of ELF3. **b, c** Pearson correlation analysis of *Spink1*/SPINK1 and *Elf3*/ELF3 mRNA expression in **b** NRasV12+Myr-AKT HCC mouse model in a stepwise manner from normal to early and advanced HCC indicated by ●, ■, ▲, ▼, ○ and □, and **c** human samples from TCGA-LIHC dataset. **d** Co-staining of SPINK1 (yellow) and ELF3 (green) protein in human HCC tumor with high and low SPINK1 expression by multiplex IHC. DAPI (blue), nucleus. Scale bar: 50 μm. **e** (Top) Computational prediction of ELF3 binding sites (S1 at -4053 to -3802 bp and S2 at -2103 to -2019 bp) on SPINK1 promoter region by Gene Transcription Regulation Database (GTRD)[33]. TSS = transcription start site. (Bottom) Reporter constructs were generated by cloning of full-length (FL) or truncated promoter fragments (pT1 to pT3) into a pGL3 vector encoding firefly luciferase. **f** Luciferase reporter assays showing relative luciferase unit (firefly/renilla) in Huh7 cells with FL or different truncations of SPINK1 (pT1 to pT3) promoters. pRL-CMV Renilla luciferase plasmid co-transfected for normalization. **g** ChIP-qPCR confirmation of ELF3 binding to predicted sites (S1 and S2) on SPINK1 promoter in Huh7 cells, by using ELF3 and IgG antibodies. **h**–**j**, **h** Relative luciferase reporter activity, **i** mRNA expression and **j** secretory SPINK1 protein levels in Huh7 cells with or without CRISPR/Cas9-mediated ELF3 knockout (shELF3-2 and shELF3-3). **k** Relative luciferase reporter activity, **l** mRNA expression, and **m** secretory SPINK1 protein levels in MHCC97L cells with or without shELF3-2 and shELF3-3, and treated with 5-FU, cisplatin or DMSO. **b** $n = 2$-3 mice; **c** $n = 371$; **d** $n = 2$ human HCC samples with high SPINK1, 2 human HCC samples with low SPINK1; **f**–**j** $n = 3$ independent experiments; **k, l** $n = 4$ independent experiments; **m** $n = 3$ independent experiments. Data were expressed as mean ± s.e.m. Statistical analysis: **b**, **c** two-sided Pearson correlation analysis, (**f**, **h**–**j**) one-way ANOVA, or (**g**, **k**–**m**) two-way ANOVA. Source data are provided as a Source Data file.

## Animal experiments

All mice (Mus musculus) were housed in Association for Assessment and Accreditation of Laboratory Animal Care International (AAALAC)-credited facility in 12 hours light/dark cycle (07:00–19:00 light, 19:00–07:00 dark), with controlled room temperature ($23 \pm 2°C$) and humidity (30-70%), in groups according to stocking density as recommended in the guide. According to the CULTAR guidelines, the diameter of a single tumor should not exceed 15 mm in mice for therapeutic studies, the decrease of body weight should not exceed 20% from baseline, and the mouse did not exhibit signs of being moribund, unconscious, or comatose, nor display a prolonged or irreversible inability to eat or drink. To ensure compliance with these guidelines, tumor volume and weight were monitored every other day during the study. At the endpoint, animals were euthanized using cervical dislocation under anesthesia as approved by CULTAR. Male mice were exclusively utilized in all animal experiments of this study, as males exhibit a significantly higher incidence rate of HCC compared to females clinically.

## Liver regeneration mouse model by DDC diet treatment

4-week-old male C57BL/6 mice were fed with 0.1% DDC in a standard diet (TestDiet) *ad libitum* 2 for weeks and sacrificed at the end of treatment[55].

## Liver regeneration mouse model by partial hepatectomy

Surgery was performed as previously described[4]. Briefly, male nude mice (BALB/AnN-nu) aged between 4 and 8 weeks were anesthetized with 80 mg/kg ketamine and 10 mg/kg xylazine by intraperitoneal (i.p.) injection. The left and median lobes were ligated separately and resected to achieve 2/3 hepatectomy. Mice were sacrificed on day 0 or day 7 after surgery.

## Diethylnitrosamine (DEN) and carbon tetrachloride (CCl₄) fibrosis-induced HCC mouse model

14-day-old B6C3F1 mice were treated with a single dose of DEN (i.p., 1 mg/kg) and then with $CCl_4$ (i.p., 0.2 mL/kg) twice weekly, starting at 8-weeks of age for 12 to 16 weeks[56]. Tumors started to form at approximately the age of 20 weeks and the humane endpoint was at the age of 24 weeks[8].

## Nonalcoholic steatohepatitis (NASH)-HCC mouse model

Male C57BL/6 wild-type littermates (8 weeks old) were fed with normal chow (NC), high fat low cholesterol diet (HFLC) or high fat high cholesterol diet (HFHC) (Specialty Feeds) *ad libitum* for 14 months. Mice fed with high-fat-high-cholesterol (HFHC) developed NASH-HCC, while HFLC diet induced only simple steatosis.

## Hydrodynamic tail vein injection (HTVI) of NRasV12 and myr-AKT1 HCC (NRAS + AKT) mouse model

6- to 8-weeks old male C57BL/6 mice were used and the procedure was performed as previously described[8]. In brief, 20 μg of plasmids encoding human AKT1 (myr-AKT1) and human neuroblastoma Ras viral oncogene homolog (NRasV12) along with sleeping beauty (SB) transposase, at a ratio of 25:1, was mixed and diluted in 2 mL sterile 0.9% sodium chloride solution. A volume corresponding to 10% of the body weight was injected through the lateral tail vein in 5 to 7 seconds. HCC tumors started to grow at 2-3 weeks post-HTVI and the humane endpoint was at approximately 6-7 weeks post-HTVI.

## Genetic depletion of CD133/Prom1 using conditional CreER-induced diphtheria toxin (DTA) expression mouse model

Prom1[C-L] C57BL/6 mice (017743)[57] generated by Richard Gilbertson and obtained from the Jackson Laboratory were crossed with the Rosa26[tdTomato] (007905) or Rosa26[DTA] (006331) C57BL/6 mice. The 4-week-old Prom1[C-L/+]; Rosa26[tdTomato/+] (Ctrl) and Prom1[C-L/+]; Rosa26[DTA/+] (Prom1-DTA) male C57BL/6 mice were administered with 7 doses of 4 mg tamoxifen (Sigma-Aldrich) or vehicle (olive oil) orally every other day, as previously described[8].

## In vivo limiting-dilution and serial transplantation assays

4- to 6-weeks old male NOD/SCID (NOD.Cg-*Prkdc^{scid} IL2rg^{tm1wjl}*/SzJ0) mice were injected subcutaneously with 5000, 10000, or 50000 Huh7 cells stably transduced with lentivirus containing non-target scrambled control (NTC) or shRNA clones targeted for SPINK1 knockdown (clones SH2 and SH3). For secondary implantation, tumors were dissociated from each group for subsequent passage into secondary mouse recipients for in vivo limiting dilution analysis or for ex vivo limiting dilution spheroid assay. Tumor incidence, tumor latency, and tumor-free survival were recorded. Tumor-initiating frequency was calculated using extreme-limiting-dilution analysis.

## NRasV12+Myr-AKT HTVI HCC proof-of-concept therapeutic mouse model

HTVI of NRasV12+Myr-AKT in 6- to 8-weeks old male C57BL/6 mice is described above. Two weeks post-HTVI, mice were separated into 2 groups and administered with $5 \times 10^7$ transducing units of lentiviruses encoding either non-target scrambled control (shNTC) or shRNA targeted at SPINK1 (shSpink1), resuspended in 100 μL PBS, via tail vein injection. Sequences of shNTC and shSpink1 are detailed in Supplementary Table S1. Sequences were cloned into the LV3 vector at the service of Shanghai GenePharma Co. At 3 weeks post-HTVI, each group of mice was further separated into 2 groups and treated with 7 doses of DMSO or 5-FU (i.p., 20 mg/kg) every other day for two weeks. At 3.5 weeks post-HTVI, mice were administered another dose of the shNTC or shSpink1 lentiviruses.

## Flow cytometry of CD133 expression in HCC mouse model

Tumor cells harvested from HCC mouse models were stained with LIVE/DEAD™ Fixable Near-IR Dead Cell Stain Kit (Invitrogen) and then with APC-conjugated CD45 antibody (clone 30-F11) (1:100, eBioscience, 17-0451-83) and APC-conjugated TER-119 antibody (clone

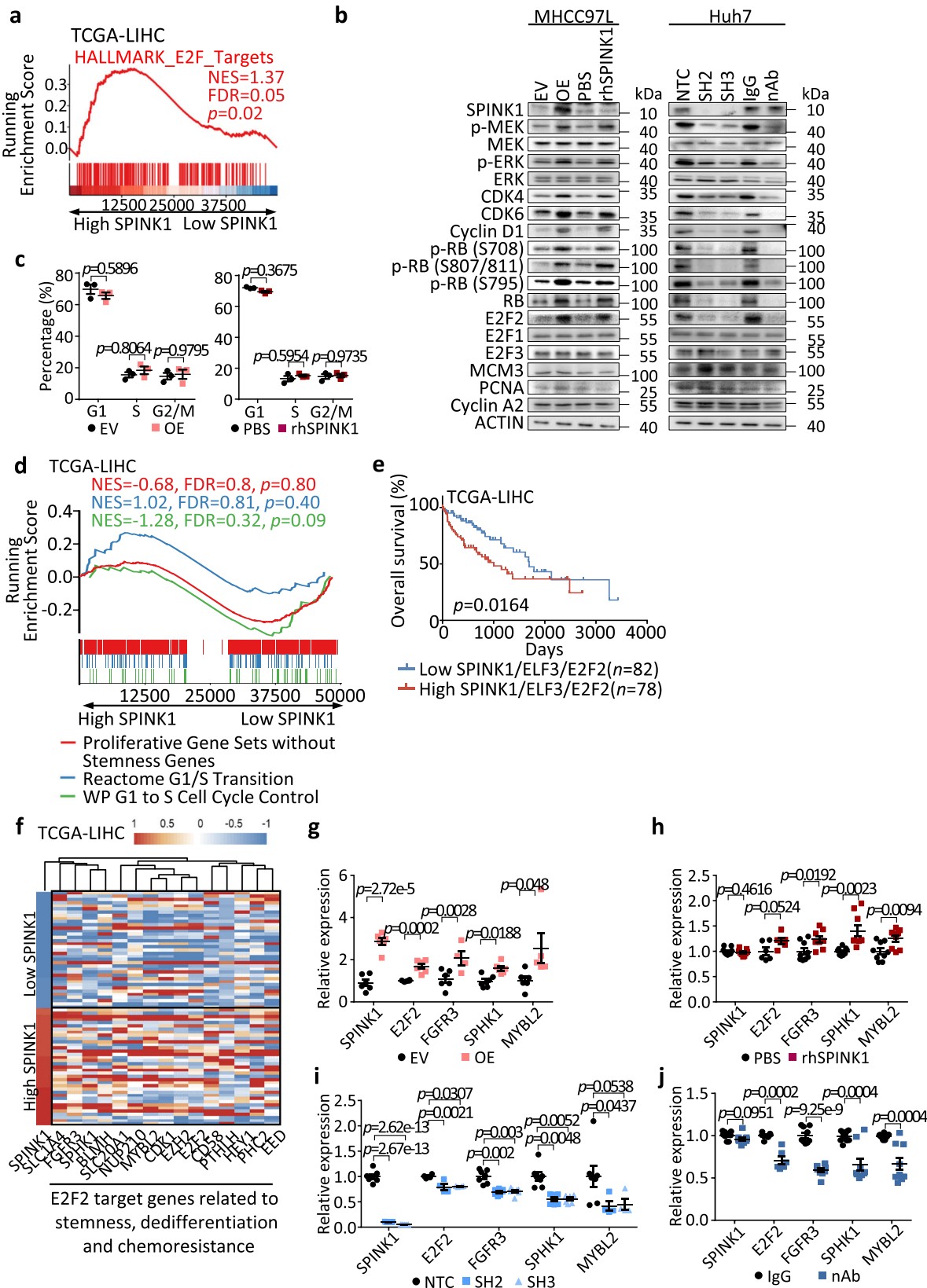

TER-119) (1:100, eBioscience, 17-5921-83). Cells were then stained with PE-conjugated CD133 (clone AC133) (1:100, Miltenyi Biotec, 130-080-801) and its respective isotype control. Cells were analyzed on a NovoCyte Advanteon (Agilent) with data processed on FlowJo (Tree Star).

**Immunohistochemistry**

After dewaxing and rehydration of the paraffin-embedded sections, the antigen retrieval was carried out using Envision Flex Target Retrieval Solution of either high or low pH (DAKO) by microwave. The section was then blocked with a Peroxidase-Blocking Solution (DAKO) and protein

**Fig. 8 | SPINK1 promotes self-renewal, dedifferentiation, chemoresistance and tumor initiation through a deregulated ERK-CDK4/6-E2F2 regulatory axis.** **a** E2F targets signature in HALLMARK was enriched in patients with high *SPINK1* expression of TCGA-LIHC cohort, via GSEA. **b** Western blot analysis for expression of SPINK1, phosphorylated and total MEK, phosphorylated and total ERK, CDK4, CDK6, Cyclin D1, phosphorylated and total Rb, E2F2, E2F1, E2F3, MCM3, PCNA and Cyclin A2 in MHCC97L cells with EV versus OE and PBS versus rhSPINK1, and in Huh7 cells with NTC versus SH2 versus SH3 and IgG versus nAb. **c** Cell cycle analysis of MHCC97L cells (left) with EV or OE, (right) with or without rhSPINK1 by flow cytometry. **d** GSEA on HCC patients segregated into high/low *SPINK1* expression using Reactome G1/S Transition, WP G1 to S Cell Cycle Control, and Proliferative Gene Sets without Stemness Genes signatures in TCGA-LIHC cohort. **e** Kaplan-Meier curve showing the percentage of overall survival in HCC patients from TCGA-

LIHC cohort with high and low *SPINK1/ELF3/E2F2* expression. **f** Heatmap showing the clustering of tumor *SPINK1* expression level of TCGA-LIHC cohort with E2F2 target genes related to stemness, dedifferentiation and chemoresistance. **g–j** qPCR analysis of the expression of *SPINK1*, and E2F2 target genes, *FGFR3*, *SPHK1* and *MYBL2* in MHCC97L cells with **g** EV versus OE and **h** PBS versus rhSPINK1, and in Huh7 cells with **i** NTC versus SH2 and SH3 and **j** IgG versus SPINK1 nAb. **b** $n = 3$ independent experiments; **c** $n = 3$ independent experiments; **g** $n = 6$ independent experiments; **h** $n = 9$ independent experiments; **i** $n = 7$ independent experiments; **j** $n = 9$ independent experiments. Data were expressed as mean ± s.e.m. Statistical analysis: **a, d** one-sided Kolmogorov–Smirnov test with adjustments for FDR, **c, g–j** two-way ANOVA, **e** log-rank test. Source data are provided as a Source Data file.

---

block (DAKO). Sections were subsequently incubated with ELF3 antibody (1:500, Novus Biologicals, NBP1-30873) overnight at 4°C. The signal was detected by incubation with HRP Labeled Polymer Anti-Rabbit (K4003, DAKO) for 30 minutes at room temperature, followed by DAB+ Substrate-Chromogen System (DAKO) and counterstained with Mayer's hematoxylin. The sections were imaged by a Zeiss Axioplan upright microscope equipped with Leica DMC6200 color digital camera.

### Multiplex immunohistochemistry (IHC)

After dewaxing and rehydration of the paraffin-embedded sections, antigen retrieval was performed using Envision Flex Target Retrieval Solution of either high or low pH (DAKO). Multiplex IHC was then performed by using Opal 7-Color Manual IHC Kit (AKOYA) according to the manufacturer's instructions. In brief, the sections were then blocked with Opal Antibody Diluent/Block (AKOYA), followed by incubation with primary antibody for either 1 hour at room temperature or overnight at 4 °C. The signal was then detected by incubation with Opal Polymer horseradish peroxidase (HRP) Ms plus Rb for 30 minutes at room temperature, followed by Opal 570 Reagent. Another two rounds of staining were repeated with Opal 520 Reagent and Opal 690 Reagent. DAPI was used to counterstain the nuclei. Primary antibodies used included SPINK1 (clone 4D4) (1:1000, Abnova, H00006690-M01), CD133 (1:100, Abcam, ab19898), ELF3 (1:500, NBP1-30873, Novus Biologicals), EGFR (1:200, 4267, Cell Signaling Technology), CD45 (1:100, Abcam, ab10558), CD3 (clone SP7) (1:100, Abcam, ab16669), α-SMA (1:100, Abcam, ab5694) and CD31 (1:100, Abcam, ab28364). The sections were imaged by Vectra Polaris imaging system (Perkin Elmer) and analyzed by Phenochart (Perkin Elmer) and inForm (Perkin Elmer).

### RNAScope

After dewaxing and rehydration of the paraffin-embedded sections, the sections were treated with RNAScope® Hydrogen Peroxide (Advanced Cell Diagnostics). Target retrieval was carried out using RNAScope® Target Retrieval (Advanced Cell Diagnostics), followed by protease treatment using Protease Plus (Advanced Cell Diagnostics). RNAScope was performed using RNAScope® 2.5 HD Reagent Kit-BROWN (Advanced Cell Diagnostics) or RNAScope® Multiplex Fluorescent Reagent Kit v2 (Advanced Cell Diagnostics) with RNA Protein Co Detection Assays (Advanced Cell Diagnostics) according to manufacturer's instructions. In brief, the probe to *Spink1* (NM_009258.5) was applied to the sections for 2 hours at 40°C in a HybEZ™ oven and then Hybridize Amp 1-6 was added accordingly. Signal was detected by applying a mix of equal volumes of BROWN-A and BROWN-B. Counterstain was performed with hematoxylin and ammonia water. Sections were imaged by a Zeiss Axioplan upright microscope equipped with Leica DMC6200 color digital camera or Vectra Polaris imaging system (Perkin Elmer) and analyzed by Phenochart (Perkin Elmer) and inForm (Perkin Elmer).

### ELISA

A total of $2-3 \times 10^5$ HCC cells were seeded in 6 well-plates. Following a two-day incubation at 37 °C, the culture medium was harvested and subsequently centrifuged at 300 g for 5 min. The supernatant was then collected for ELISA. Measurements of secretory SPINK1 were carried out using the human SPINK1 DuoSet ELISA (DY7496-05, R&D Systems) according to the manufacturer's protocol. The signal was measured at a wavelength of 450 nm using the VICTOR3 1420 Multilabel Plate Counter (Perkin Elmer).

### Lentiviral production and cell transduction

Human-specific shRNA sequence cloned in pLKO.1 were purchased from Guangzhou IGE Biotechnology Ltd. shRNA sequences targeted at human SPINK1 (NM_001354966.1), ELF3 (NM_ 004433) and EGFR (NM_005228) are detailed in Supplementary Table S1. Sequences were transfected into 293FT cells and then packaged using MISSION Lentiviral Packaging Mix (Sigma-Aldrich). The mRNA sequence of SPINK1 (NM_001354966.1) cloned into pDONR™221 was purchased from HITRO BioTech and was shuttled from pDONR™221 to pEZ-Lv199 (GeneCopoeia) through a LR reaction. pEZ-Lv199 was used as empty vector control. The sequence was transfected into 293 T/17 cells and then packaged using Lenti-Pac HIV expression packaging kit (Gene-Copoeia). Virus-containing supernatants were collected for transduction to generate HCC cells with stable SPINK1, ELF3 or EGFR repression or SPINK1 overexpression. Puromycin or blasticidin was used for cell selection.

### RNA extraction, cDNA synthesis and qPCR

Total RNA was extracted by RNAisoPlus (Takara) and cDNA was synthesized with PrimeScript RT Master Mix (Takara). qPCR was performed using EvaGreen qPCR Master Mix (Applied Biological Materials) on a LightCycler 480 Instrument II analyzer (Roche) with data analyzed by the LightCycler 480 Instrument II software (Roche). Relative expression differences were calculated using the $2^{-\Delta\Delta Ct}$ method. Primer sequences are detailed in Supplementary Table S2.

### Limiting dilution assay

Cells at limited dilutions were cultured in 100 μL serum-free DMEM medium supplemented with B27 (1:50; Invitrogen), 20 ng/mL human recombinant EGF (Sigma-Aldrich), 10 ng/mL human recombinant basic FGF (Sigma-Aldrich), 4 mg/mL insulin (Sigma-Aldrich), 500 U/mL penicillin, 500 mg/mL streptomycin (Invitrogen) and 0.25% methylcellulose (Sigma-Aldrich) in polyHEMA-coated 96-well plates. 30 μL of medium was supplemented to each well every other day. The number of wells with sphere formation was counted at 7-14 days. Tumor-initiating cell frequency was calculated using extreme limiting dilution analysis.

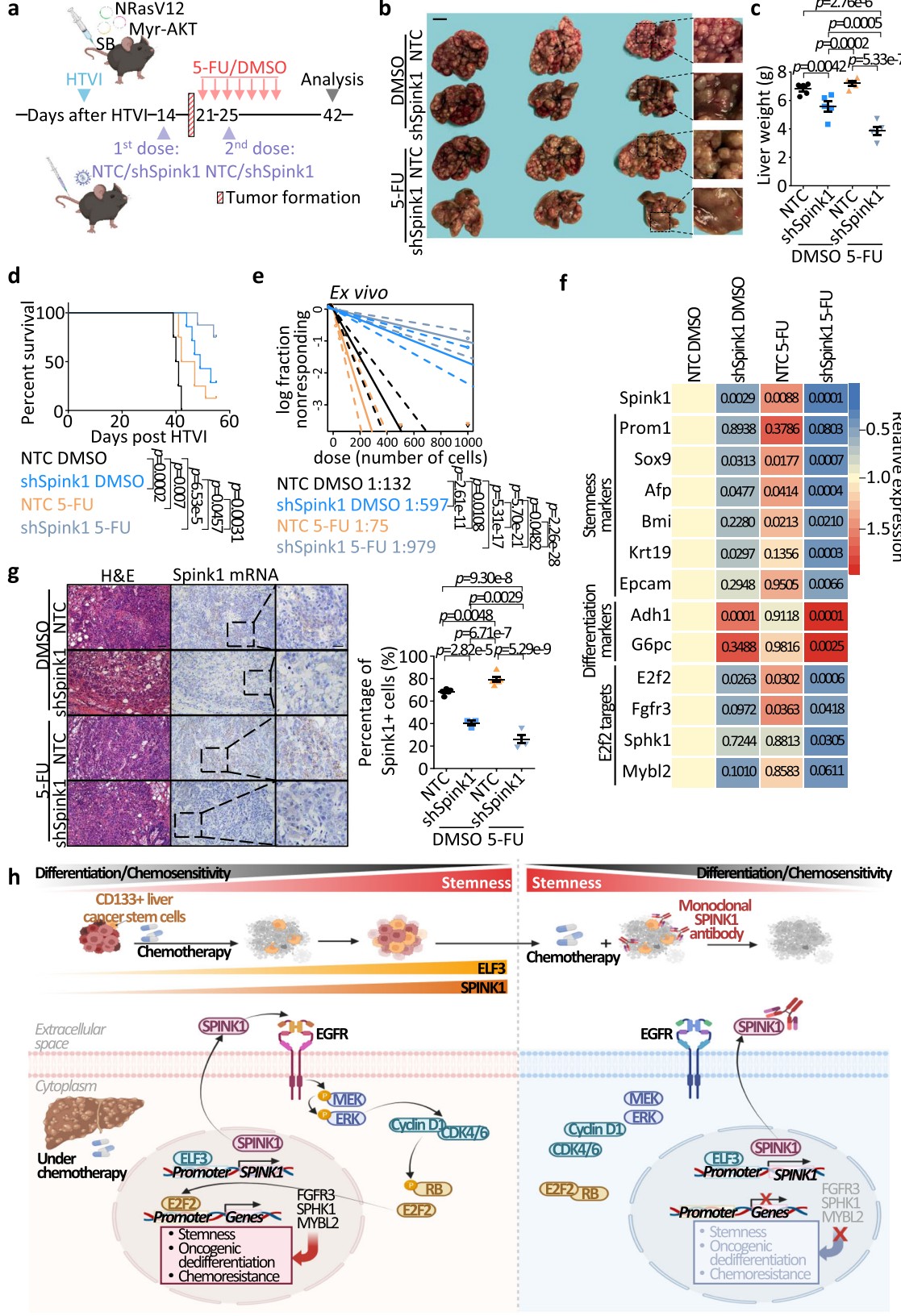

## Annexin V apoptosis assay

Following a 48-hour treatment with 5-FU or cisplatin, cells were harvested and stained with propidium iodide (PI) (BioLegend) and FITC-conjugated Annexin V (BioVision) for 30 minutes on ice. Samples were then analyzed on BD LSRFortessa cell analyzer (BD Biosciences) with data processed on BD FACSDiva (version 8.0.1) and FlowJo (Tree Star).

## Cell cycle analysis

Cells were serum-starved for 48 hours and then cultured in complete medium, supplemented with 10% FBS. Cells were then collected and fixed in 70% ice-cold ethanol overnight. Cells were stained with propidium iodide (PI) (BioLegend) for 30 minutes on ice and analyzed on a NovoCyte Advanteon (Agilent) with data processed on FlowJo (Tree Star).

**Fig. 9 | Therapeutic targeting of SPINK1 may represent a promising treatment option for chemoresistant HCC. a** Experimental scheme of Spink1 knockdown in combination with 5-FU treatment in NRasV12+Myr-AKT HTVI HCC model. Lentiviral particles with NTC or encoded Spink1 knockdown (shSpink1) were administered twice at 2- and 3.5-week post HTVI. After tumor initiation, mice were treated with 5-FU (i.p., 20 mg/kg) or DMSO every other day for two weeks. Liver tissues were collected 7-week post HTVI. **b** Representative images and **c** quantitative data of liver weights from NTC or shSpink1 mice treated with DMSO or 5-FU. Scale bar = 1 cm. **d** Kaplan–Meier survival curve showing survival percentage of each annotated group. **e** Ex vivo limiting dilution analysis for frequency of TICs in cells harvested from NTC or shSpink1 mice with DMSO or 5-FU treatments. **f** qPCR analysis of the expression of *Spink1*, stemness and differentiation-related genes, *E2F2* and E2F2

target genes. *p*-values from comparisons to NTC DMSO group were shown in the corresponding boxes. **g** (Top) Representative image and (bottom) quantitative data of liver tissues from NTC or shSpink1 mice treated with DMSO or 5-FU with *Spink1* mRNA staining by RNAScope. Scale bar in low magnification = 50 μm, high magnification = 20 μm. **h** Cartoon summary of findings. **a** $n = 14–15$ mice; **b**, **c** $n = 5$ mice; **d** $n = 7$-8 mice; **e** 19–20 replicates; **f** $n = 10$–18 independent experiments; **g** $n = 5$ views for Ctrl DMSO group, 3 views for shSpink1 DMSO group, 5 views for Ctrl 5-FU group, 4 views for shSpink1 5-FU group. Data were expressed as mean ± s.e.m. Statistical analysis: (**c**, **f**–**g**) two-way ANOVA, **d** log-rank test, or **e** one-sided Person's χ2 test with 95% confidence intervals. Source data are provided as a Source Data file. Illustration for (**a**, **h**) was created using BioRender.com.

## Identification of transcription factor regulating SPINK1

Gene transcription factors binding to SPINK1 promoter was predicted from a public collection of ChIP-seq data website Gene Transcription Regulation Database (GTRD) (https://gtrd.biouml.org)[33]. Correlation of SPINK1 with the candidate gene transcription factors was calculated using TCGA-LIHC data. Binding sites were predicted using GTRD.

## Luciferase reporter assay

The full-length SPINK1 promoter was amplified from the HCC cell line Huh7 and cloned into the pGL3-Basic luciferase reporter vector (Promega) using the primers (F: 5′-AAAGAGCTCGGGCAACCCT TTCTTTGTGG-3′; R: 5′-AAAACGCGTCCGCACTTACCACGTCTCTT-3′). pT1 (ΔS2), pT2 (ΔS2) and pT3 (ΔS1 + ΔS2) were purchased from Guangzhou IGE Biotechnology Ltd. Luciferase reporter constructs and pRL-CMV *Renilla* luciferase control were transiently co-transfected into HCC cell lines by Lipofectamine 3000 reagent (Thermo Scientific). Luciferase activity was determined by the Dual-Glo Luciferase Assay System (Promega) and detected by Victor[3] 1420 Multilabel Plate Counter (PerkinElmer). Luciferase reporter activity represented by a ratio of firefly: *Renilla* luminescence.

## Chromatin immunoprecipitation (ChIP) assay

ChIP was performed with the Magna ChIP G-Chromatin Immunoprecipitation Kit (Millipore). In brief, cells were crosslinked by 1% formaldehyde. DNA was then sonicated by Bioruptor® Pico sonication system (Diagenode) and subsequently immunoprecipitated with anti-ELF3 (1:100, Novus Biologicals, NBP1-30873) or rabbit IgG control (1:100, Bethyl Laboratories, P120-101). Immunoprecipitated and eluted DNA was purified with columns and amplified by qPCR with the following primers: S1 – (F: 5′-CAACAGGTGCCAGCCCAATA-3′), (R: 5′-GAAATCCTGCCACCGTGCTA-3′) and S2 - (F: 5′-GGAGCCAAGTCATA-CAGGACC-3′), (R: 5′-ATCCTTTCCCCCTGGGTTTC-3′).

## Co-immunoprecipitation

Cells were washed with ice-cold PBS and then lysed with ice-cold NETN buffer (20 mM Tris-HCl pH8.0, 100 mM NaCl, 1 mM EDTA and 0.5% v/v NP40). 1 μg lysate was incubated with SPINK1 antibody (clone 4D4) (1:100, Abnova, H00006690-M01) or EGFR antibody (clone D38B1) (1:100, Cell Signaling Technology, #4267) with gentle shaking overnight at 4°C. Mouse IgG (14-4714-82, eBioscience) and rabbit IgG (P120-101, Bethyl Laboratories) were used as control. The lysate was washed with ice-cold NETN buffer twice and then incubated with Protein A Sepharose beads (BioVision) with gentle shaking for 3 hours at 4°C. The supernatant was collected after centrifugation and then analyzed by Western blot.

## Western blot

Protein lysates were quantified and resolved on an SDS-PAGE gel followed by transferring onto a PVDF membrane (Pall). The membrane was then immunoblotted with primary and secondary antibodies.

Signal was detected by ECL™ Select Western blotting Detection Reagent (Cytiva). The following antibodies was used: SPINK1 (clone 4D4) (1:1000, Abnova, H00006690-M01), ELF3 (1:1000, Novus Biologicals, NBP1-30873), EGFR (1:1000, Cell Signaling Technology, #2232), p-MEK1/2 (Ser217/221) (1:1000, Cell Signaling Technology, #9121), MEK1/2 (1:1000, Cell Signaling Technology, #9122), p-ERK1/2 (Thr202/Tyr204) (1:1000, Cell Signaling Technology, #9101), ERK1/2 (1:1000, Cell Signaling Technology, #9102), CDK4 (clone D9G3E) (1:1000, Cell Signaling Technology, #12790), CDK6 (clone DCS83) (1:1000, Cell Signaling Technology, #3136), cyclin D1 (clone 92G2) (1:1000, Cell Signaling Technology, #2978), p-RB (Ser708) (clone D59B7) (1:1000, Cell Signaling Technology, #8180), p-RB (Ser795) (1:1000, Cell Signaling Technology, #9301), p-RB (Ser807/811) (clone D20B12) (1:1000, Cell Signaling Technology, #8516), Rb (clone 4H1) (1:2000, Cell Signaling Technology, #9309), E2F2 (clone TFE-25) (1:500, Santa Cruz, sc-9967), E2F1 (clone KH95) (1:500, Santa Cruz, sc-251), E2F3 (clone PG30) (1:500, Santa Cruz, sc-56665), MCM3 (clone D47B6) (1:1000, Cell Signaling Technology, #4003), PCNA (clone PC10) (1:1000, Abcam, ab29), Cyclin A2 (clone E1D9T) (1:1000, Cell Signaling Technology, #91500) and β-actin (clone AC-74) (1:5000, Sigma-Aldrich, A5316). Images were captured using BioRad ImageLab Touch software (version 2.4.0.03).

## Reagents

5-fluorouracil, tamoxifen, DEN and $CCl_4$ were purchased from Sigma-Aldrich. Cisplatin was purchased from Hansoh Pharma. Erlotinib (OSI-774) was purchased from Selleckchem and used at a concentration of 10 μM. Palbociclib (P-7744) was purchased from LC Laboratories and used at a concentration of 10 ng/mL. Human SPINK1 recombinant protein (H00006690-P01) was purchased from Abnova and used at a concentration of 100 ng/mL. Normal IgG control was purchased from R&D Systems. SPINK1 neutralizing antibody (clone 839304) (MAB74961) was purchased from R&D Systems and used at a concentration of 1 μg/mL.

## Statistical analysis

All statistical analyzes were performed using GraphPad Prism 6. Two-sided Unpaired Student's *t*-tests were used for comparison between two independent groups and two-sided Paired Student's *t*-test was used for paired data. Two-sided Ordinary one-way ANOVA with Tukey's multiple comparison test was used for analysis for more than two groups and two-sided two-way ANOVA with Sidak's multiple comparison was used for analysis for more than two factors. log-rank test was used in Kaplan-Meier survival curves. One-sided Kolmogorov-Smirnov test with adjustments for FDR was used for gene set enrichment analysis (GSEA). Data are shown as the mean ± SEM (standard error of mean). Comparison with *p* value less than 0.05 was regarded as statistically significant.

## Reporting summary

Further information on research design is available in the Nature Portfolio Reporting Summary linked to this article.

## Data availability

The TCGA-LIHC publicly available data used in this study are available in the dbGaP repository under accession phs000178.v11.p8. The transcriptome sequencing data of sorted CD133+ and CD133- cells from DEN+CCl$_4$ HCC model, NRasV12+Myr AKT HCC model and liver regeneration model generated in this study has been deposited in the European Nucleotide Archive (ENA) database under accession code PRJEB59278. The transcriptome sequencing data of Prom1-DTA mouse models are deposited in the GEO database under accession code GSE181515. The scRNA-seq data profiles human liver development is obtained from http://collections.cellatlas.io/liver-development. The scRNA-seq data of human HCC is publicly available in the GEO database under accession code GSE151530. The remaining data are available within the Article, Supplementary Information or Source Data file. Source data are provided with this paper.

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

## Acknowledgements

This project is supported in part by grants from the Research Grants Council of Hong Kong – Collaborative Research Fund (C7026-18G), Research Grants Council of Hong Kong – Research Fellow Scheme (RFS2122-7S05), Croucher Foundation – Croucher Senior Research Fellowship, and Health and Medical Research Fund (09202046). We also acknowledge the funding support from the "Laboratory for Synthetic Chemistry and Chemical Biology" and the "Centre for Translational and Stem Cell Biology" under the Health@InnoHK Program launched by the Innovation and Technology Commission, The Government of Hong Kong Special Administrative Region of the People's Republic of China, Guangdong Science and Technology Department (2020B1212030004) and National Natural Science Foundation of China (82373080). We thank the Centre for PanorOmic Sciences (The University of Hong Kong) for providing and maintaining the equipment and technical support needed for flow cytometry, animal imaging and confocal microscopy studies. We thank the Centre for Comparative Medicine Research (The University of Hong Kong) for supporting our animal work studies. The results published here are in part based upon data generated by The Cancer Genome Atlas (TCGA) Research Network: http://www.cancer.gov/tcga. Publication was made possible in part by support from the HKU Libraries Open Access Author Fund sponsored by the HKU Libraries.

## Author contributions

K.F.M., L.Z. and S.M. conceived the project and designed the studies. K.F.M. and L.Z. performed the research and analyzed and interpreted the data with the help of H.Y., K.H.L. and Y.T.L. L.Z. generated the liver regeneration mouse model, as well as the HCC and Prom1-DTA mouse models and prepared cells for transcriptome sequencing. H.Y. and K.H.L. aided in the proof-of-concept therapeutic animal model. W.C. generated the data of mouse fetal liver models at different developmental stages. JPY obtained patient consent and provided the clinical samples for clinical analysis. J.Y. provided data relating to nonalcoholic fatty liver-related HCC. X.Y.G. and M.L. provided data relating to mouse and human liver development. T.L. provided Palbociclib and offered scientific advice. K.F.M., L.Z. and S.M. wrote the paper. S.M. supervised the project and provided funding for the study.

## Competing interests

The authors declare no competing interests.
