## [Peer Review File · Nature Communications]

Reviewers' Comments:

Reviewer #1:

Remarks to the Author:

In this manuscript, Man et al. explored the implication of SPINK1 in tumour plasticity in hepatocellular carcinoma (HCC) and the putative therapeutic interest to target SPINK1 for therapy. Through a collection of complementary approaches, they provided evidence that SPINK1 is associated with CD133-positive cells, a marker of tumour initiating/propagating cells. This is shown in CD133-positive cells from HCC models, developing livers, and in experimental setting of tumour dedifferentiation. They showed that increased transcriptional activity of SPINK1 is mediated by promoter binding of ELF3, which was found upregulated following chemotherapy, similarly to CD133. The link between these modulators is provided by showing reduced ELF3 and SPINK1 following CD133 depletion. Furthermore, the authors provided a possible mechanism of action by showing that SPINK1 binds to EGFR, leading to the activation of EGFR-ERK-CDK4/6-E2F2 signalling circuit, which they associated with self-renewal and chemoresistance. Finally, they provide experimental data indicating that SPINK1 blockage could be a therapeutically pertinent option for further investigation.

The manuscript is well structured and written, with relevant aspects addressed in the introduction section and pertinent points discussed in the discussion section. Findings are supported by several - convincing - experimental approaches. I have a concern about the novelty of some findings as sets of experimental evidence, nicely linked in the present manuscript and documented in the context of HCC, have been previously reported. For example, the link between SPINK1 and EGFR has been reported in several studies (e.g. PMID 26437224, 19737965, 26037168, 24619958), and including in liver cells (PMID 28845526). None of these studies have been cited by the authors. Additional results previously reported, and not cited in the present manuscript are, for example, those documenting: 1) SPINK1 as a putative biomarker for the early detection, targeting, and response prediction to Immune Checkpoint Blockage in HCC (PMID 35924241); 2) SPINK1 regulation of proliferation, migration, invasion and radiation resistance in cancer patients chemoradiotherapy (PMID 36053457).

Specific points:

- The authors used through the manuscript the terminology of "tumor initiating/propagating cells". As reported in the introduction, these cells are "less differentiated cells that are resistant to therapy and associated with the development of tumor relapse". However, it is not clear based on which cellular and molecular properties the authors can attribute this definition to the cells referred in their experimental studies. This ambiguity must be clarified (e.g. in relation to CD133,) and supported with references, to facilitate readers less familiar to this terminology, particularly for articles published in journals with a wider audience. Furthermore, experimental documentation should be provided to corroborate that cells are indeed "tumor initiating/propagating cells" or rather subsets of (resistant) cancer cells. The use of CD133 marker should be complemented with other features.
- The authors illustrated how EGFR silencing abrogates SPINK1-mediated self-renewal and chemoresistance. In view of EGFR implication on the resistance of HCC cells to RTK inhibitors like lenvatinib, the authors should assess whether EGFR inhibitors used in the clinic recapitulates the effects observed on tumour plasticity shown by SPINK1 modulation and the relation with HCC resistance.
- The authors proposed a link of the EGFR-ERK-CDK4/6-E2F2 signalling circuit with self-renewal and chemoresistance. They should determine experimentally the contribution of this circuit in self-renewal and chemoresistance by targeting, for example, MEK/ERK and CDK4/6 with available inhibitors.
- In the HTVI experiments reported in Figure 9a-d, the authors performed a lentiviral injection with shSpINK1 (or control) after 2 weeks, followed by a 5-FU treatment after 3 weeks of HTVI. In the Methods sections, the authors claimed that in the NRasV12:Myr-AKT model "tumours started to grow at 2-3 weeks post-HITV", although no experimental data provide this assumption. Additional studies, ideally adding a luciferase reporter for longitudinal bioluminescence imaging, would

contribute clarifying at which phase of tumour formation SPINK1 silencing and 5FU treatment act (before tumours are formed or when tumours are already formed). Moreover, SPINK1 silenced in shSPINK1 tumours should be documented by RT-qPCR and/or Western Blot analyses.

- Transcriptomics data with list of differentially expressed genes in CD133+ vs CD133- cells must be available through open access platforms (e.g. EBI, ...).

- Molecular weight (kDa) should be reported in all western blots shown in panels, e.g. on the right.

- The number of mice used for each study and for statistical analysis, should be reported in the corresponding figure legends.

Minor:

- Page 8 line 205: the following sentence should be corrected "We next explored delineating the receptor to which SPINK1 would bind on HCC cells to...".

Reviewer #2:

Remarks to the Author:

Thank you very much for inviting me to review the manuscript "SPINK1-induced tumor plasticity provides a therapeutic window for chemotherapy in hepatocellular carcinoma" by Man et al.

Building on their previous work, the authors investigated the role of CD133+ cells in hepatocellular carcinoma (HCC) and their response to chemotherapy. By comparing the transcriptomes of normal CD133+ liver cells and CD133+ HCC cells, they identified that SPINK1 expression was preferentially enriched in CD133+ HCC cells. Treatment with chemotherapy drugs, 5-FU and cisplatin, led to the upregulation of ELF3-mediated transcription and secretion of SPINK1, which further promoted tumor initiation, stemness, dedifferentiation, and chemoresistance in HCC cells. This effect was mediated through the binding of SPINK1 to EGFR, triggering the ERK-CDK4/6-E2F2 signaling cascade. Moreover, inhibiting SPINK1 function using neutralizing antibodies or lentivirus-mediated knockdown of Spink1 reduced HCC growth and improved chemotherapy response. Authors conclude that targeting SPINK1 could be a potential therapeutic strategy for HCC treatment.

This is a well written manuscript, the conclusions are supported by the data. Functional studies have been performed in vitro and in vivo to identify a novel mechanistic basis for chemoresistance. I have a few questions and comments:

For figures, I would encourage the authors to individually point out what each figure depicts rather than use broad strokes. For instance, the entire Figure 1 is cited in the end of the first paragraph as Fig 1b-g. Instead it would be better to cite what fig 1b or 1c are depicting in the appropriate text.

It is surprising that of the 615 genes upregulated in CD133+ mouse HCC cells, only 2 pass their screening of also being elevated in human HCC and predicting survival. Authors can maybe comment on why this might be the case.

In Figure 2i the CD133+ cells are not visualized well. Moreover, in figure 2i, Spink1 appears to be overexpressed diffusely across the tumor and is not restricted to the Prom1+ cells. However, in Figure 2b, authors show that Spink1 mRNA expression is mostly restricted to the Prom1+ cells, can they explain this?

Several prior papers have pointed out that SPINK1 is overexpressed in HCC and is associated with poor prognosis (PMID: 23527199, PMID: 34595116, PMID: 32066292, PMID: 27028242) so a lot of the correlative results shown in figure 3 can potentially be moved to the supplementary data so we can focus on the functional analysis presented in later figures.

One of the key results is that "Knockdown of SPINK1 also resulted in a profound decrease in the

ability of cells to resist 5-FU and cisplatin, as indicated by enhanced rates of apoptosis in the knockout clones as compared to the control (Fig 4h)". Even though the quantitative graphs below show a significant change the representative flow images show only a 4-5% change with Spink1 knockdown. Can authors comment?

Moreover, it's not clear if Fig 4h and 4g are from in vivo or in vitro data.

In the discussion authors say "we report on the role of endogenous/secretory SPINK1 in inducing HCC to a more stem/progenitor state and sensitizing HCC cells to chemotherapy." I would say making HCC cells resistant to chemotherapy or desensitizing cells to chemotherapy.

Authors should discuss how they expect SPINK1 inhibitors to be used in HCC. Do they think they should be used as adjuvant therapy with TACE? Or in patients who don't respond to TACE?

The same authors have previously suggested that "CD133+ HCC cells contribute to chemoresistance preferential activation of Akt/PKB and Bcl-2 cell survival response." Authors should discuss if these two mechanisms are complementary or unconnected.

Reviewer #3:

Remarks to the Author:

The manuscript entitled "SPINK1-induced tumor plasticity provides a therapeutic window for chemotherapy in hepatocellular carcinoma" submitted by Man et al made attempted to show SPINK1 closely associated with CD133+ HCC, liver development, and tumor dedifferentiation. They found that enhanced transcriptional activity of SPINK1 was mediated by promoter binding of ELF3, which like CD133, increased following chemotherapy treatment. Functionally, SPINK1 overexpression enhanced tumor initiation, self-renewal, and chemoresistance, with a deregulated EGFR-ERK-CDK4/6-E2F2 signaling engaged to induce dedifferentiation of HCC cells into their ancestral lineages. They eventually proposed that targeting oncofetal SPINK1 may represent a good therapeutic option for HCC clinics. Although the authors presented several lines of interesting data to make the final conclusions, the whole study suffers from the lacking of appropriate elucidation of functional mechanisms, with the majority of the experimental data basically not supporting the hypothesis. Significant improvements are essential for the current work.

Some minors:

1. Fig. 1a and Supplementary Fig. S1a intended to show that 5-FU or cisplatin enriched for a CD133/Prom1+ liver tumor initiating/propagating cell subset. However, the flow cytometry analysis for Prom1 expression in mouse NRasV12+Myr-AKT HTVI HCC tumors with DMSO or 5-FU treatment was demonstrated, the reason using these mice (hydrodynamic tail vein injected (HTVI) NRasV12+Myr-AKT proto-oncogenes-driven HCC tumor mouse model, rather than a de novo tumorigenesis model) remains fairly unclear, should be explained in the text.
2. Page 5, the 1st paragraph. Notably, 5-FU treatment alone did not reduce tumor growth and importantly also led to an elevated stem cell frequency suggesting a higher chance of tumor recurrence if only chemotherapy is utilized (Fig. 1b-g). The sentence appears to be quite embarrassed and needs to be rewritten.
3. Page 5, the 2nd paragraph. While we and others have now demonstrated a functional role of CD133/Prom1 in HCC, CD133 unfortunately is not specific to HCC and is also expressed in the regenerating liver. Poorly organized.
4. Page 5. Upon further correlation with survival using human clinical samples extracted from TCGA-LIHC, IGFALS, SPINK1 and B4GALNT1 were identified as potential candidates (Fig. 2a and Supplementary Fig. S2a, b). The selection criteria were not provided at all. Furthermore, supplementary Fig. S2a, b are too busy and should be simplified.
5. Page 6. SPINK1 is not expressed in the normal liver nor many other major organs and immune cells, making it an attractive therapeutic target. Did the authors analyze SPINK1 expression pattern in the overall liver tumor microenvironment, rather than simply considering the tumor bulk or CD133+ cells? Chemotherapy induces SPINK1 expression in the tumor stroma, although not tumor cells, how to reconcile the differential responses?

6. Supplementary Figure S1b shows representative image of livers from NRasV12+Myr-AKT HTVI HCC mice treated with DMSO, 5-FU or cisplatin with Spink1 mRNA staining by RNAScope. However, the data need to be accompanied by protein level examinations such as immunohistochemistry staining against Spink1, to allow the conclusions make further sense.
7. Page 6. Spink1 was also found to be enriched in HCC tumors enriched for a CD133+ liver tumor initiating/propagating cell subset following either 5-FU or cisplatin treatment, why? Did the authors investigate the mechanism(s) underlying the induction or enrichment of Spink1 expression in HCC tumors? How about the stromal cells, such as fibroblasts and immune cell subtypes?
8. Just according to the data that SPINK1 associates closely with CD133, it is not appropriate to inferred that SPINK1 might be associated with liver development (Supplementary Fig. S3c). The speculation is not valid.
9. Page 7. Using CytoTRACE, a computational framework for predicting differentiation from former analysis of scRNA-seq data, the authors stated to have found high SPINK1 expression overlapping with the less differentiated hepatic cells (Fig. 3d). The opposite was indicated in fact.
10. It was explained that patients in TCGA-LIHC cohort were ranked according to the transcriptomic level of SPINK1 in their tumors. However, 10% of highest-ranking patients were marked as high SPINK1, while 10% of lowest-ranking patients were marked as low SPINK1, where was this indicated in Fig. 3f-g? Should be clearly marked through.
11. Fig. 3i. High proteomic SPINK1 levels significantly correlated with a worst overall survival. Kaplan-Meier analysis of the overall survival in HCC patients with high and low SPINK1 scoring in their tumor samples. * $p < 0.05$ by log-rank test. SPINK1 expression either low or high, with the lines corresponding to each status basically missing (had better follow the presentation manner of Fig. 8a).
12. Fig. 3j shows the transcriptomic level of SPINK1 in tumors (T) and paired adjacent non-tumoral livers (NT) of NASH related HCCs. How many patients' data were acquired and assessed? Should be well stated.
13. Fig. 6a. Coimmunoprecipitation analysis for validation of EGFR as an interacting protein partner of SPINK1 in Huh7 cells. The data are incomplete, as input samples need to be fully provided to cover all the 4 lanes of lysates, rather than simply showing the input of SPINK1, EGFR and actin in only one lane.
14. Fig. 6c, d show data from in vitro limiting dilution analysis for frequency of TICs (c) MHCC97L with rhSPINK1 treatment or (d) SPINK1 overexpression in the presence or absence of shRNA against EGFR stably transduced. The authors had better display the growth curves or proliferative potential of HCC cells to allow a more straightforward comparison.
15. Supplementary Fig. S4a indicates the western blot data of SPINK1 regulation through an EGFR axis, with SPINK1 overexpressed in MHCC97L cells with EGFR knockdown. The quality of WB data are generally poor, need to be redone.
16. Page 9. To seek hints into the possible upstream regulatory element that leads to SPINK1 overexpression in HCC, gene transcription factors binding to SPINK1 promoter was predicted from a public collection of ChIP-seq data website - Gene Transcription Regulation Database (GTRD). Grammar problems throughout the sentence.
17. Supplementary Fig. S4g, lacks the expression data of SPINK1.
18. Page 10. As E2F is widely known to be linked to cell cycle regulation, we first examined if SPINK1 may promote HCC through altering cell cycle proliferation. Sick sentence, needs to be rewritten.
19. Fig. 8b. Western blot shows the expression of SPINK1, phosphorylated and total MEK, 958 phosphorylated and total ERK, CDK4, CDK6, Cyclin D1, phosphorylated and total Rb, E2F2, etc. The levels of PCNA and MCM3 need to be examined, with Cyclin A2 to be analyzed in parallel.
20. Fig. 8d shows GSEA data on HCC patients segregated into high/low SPINK1 expression using Reactome G1/S Transition, WP G1 to S Cell Cycle Control, and proliferative gene sets without stemness genes signatures in TCGA-LIHC cohort. The GSEA graphs are too small and had better be enlarged to allow visualization.
21. The manuscript demonstrated that 5-FU or cisplatin chemotherapy would functionally enrich for CD133/Prom1, as a side effect of treatments, suggesting that while CD133 may be proliferative, they may also be driven by other mechanisms. SPINK1-mediated tumor plasticity is only one of such potential mechanisms, the authors need to integrate into discussion.

18 September 2023

RE: Nature Communications NCOMMS-23-16086

SPINK1-induced tumor plasticity provides a therapeutic window for chemotherapy in hepatocellular carcinoma

REVIEWER COMMENTS

Reviewer #1 - HCC mouse models, stemness, signaling:

In this manuscript, Man et al. explored the implication of SPINK1 in tumour plasticity in hepatocellular carcinoma (HCC) and the putative therapeutic interest to target SPINK1 for therapy. Through a collection of complementary approaches, they provided evidence that SPINK1 is associated with CD133-positive cells, a marker of tumour initiating/propagating cells. This is shown in CD133-positive cells from HCC models, developing livers, and in experimental setting of tumour dedifferentiation. They showed that increased transcriptional activity of SPINK1 is mediated by promoter binding of ELF3, which was found upregulated following chemotherapy, similarly to CD133. The link between these modulators is provided by showing reduced ELF3 and SPINK1 following CD133 depletion. Furthermore, the authors provided a possible mechanism of action by showing that SPINK1 binds to EGFR, leading to the activation of EGFR-ERK-CDK4/6-E2F2 signalling circuit, which they associated with self-renewal and chemoresistance. Finally, they provide experimental data indicating that SPINK1 blockage could be a therapeutically pertinent option for further investigation.

The manuscript is well structured and written, with relevant aspects addressed in the introduction section and pertinent points discussed in the discussion section. Findings are supported by several - convincing - experimental approaches. I have a concern about the novelty of some finding as sets of experimental evidence, nicely linked in the present manuscript and documented in the context of HCC, have been previously reported. For example, the link between SPINK1 and EGFR has been reported in several studies (e.g. PMID 26437224, 19737965, 26037168, 24619958), and including in liver cells (PMID 28845526). None of these studies have been cited by the authors. Additional results previously reported, and not cited in the present manuscript are, for example, those documenting: 1) SPINK1 as a putative biomarker for the early detection, targeting, and response prediction to Immune Checkpoint Blockage in HCC (PMID 35924241); 2) SPINK1 regulation of proliferation, migration, invasion and radiation resistance in cancer patients chemoradiotherapy (PMID 36053457).

Reply: Thank you for your positive feedback and thoughtful comments on our manuscript. We appreciate you taking the time to provide this constructive critique to improve our work. You have raised a fair point about the novelty of some of our findings. We agree that elements of our results have been reported previously in HCC and other cancer types and we should acknowledge these relevant prior studies. While the binding of SPINK1 to EGFR signaling was previously shown in pancreatic cancer (PMID: 24619958), this interaction had not been demonstrated previously in HCC cells. By providing evidence of this mechanism in the HCC context, our study expands the understanding of SPINK1's role beyond other cancer types. Furthermore, while previous studies reported a link between SPINK1 and chemoresistance in HCC (PMID: 36053457), our work provides novel mechanistic insights into how SPINK1 specifically promotes chemoresistance in HCC through induction of tumor plasticity. We demonstrate for the first time that chemotherapy upregulates SPINK1 via ELF3, where SPINK1 would subsequently enhance self-renewal,

dedifferentiation and the cancer stem cell phenotype through activation of EGFR-ERK-CDK4/6-Cyclin D1-E2F2 signaling pathway, conferring resistance to treatment. No prior studies had connected SPINK1 expression to tumor plasticity processes that underlie chemoresistance in HCC. Therefore, we believe our findings advance understanding of the molecular mechanisms through which SPINK1 modulates therapeutic response in HCC and is novel in this way. The references are now incorporated in pages 16 and 17 in our revised manuscript.

Specific points:

- The authors used through the manuscript the terminology of “tumor initiating/propagating cells”. As reported in the introduction, these cells are “less differentiated cells that are resistant to therapy and associated with the development of tumor relapse”. However, it is not clear based on which cellular and molecular properties the authors can attribute this definition to the cells referred in their experimental studies. This ambiguity must be clarified (e.g. in relation to CD133, ...) and supported with references, to facilitate readers less familiar to this terminology, particularly for articles published in journals with a wider audience. Furthermore, experimental documentation should be provided to corroborate that cells are indeed “tumor initiating/propagating cells” or rather subsets of (resistant) cancer cells. The use of CD133 marker should be complemented with other features.

Reply: Thank you for the comment. We do feel that a better explanation on how tumor-initiating cells is defined is indeed needed, in particular for a journal like Nature Communications with a broader audience. To date, there has been controversy regarding identification of CSCs primarily due to the technical differences in experimentation and CSC assays, etc. (PMID: 36651074). How tumor-initiating cells is defined or supported in this manuscript is by various means, including (1) correlation with CD133 which is a widely reported surrogate marker of liver cancer stem cells (PMID: 17570225; 21112564; 34588223), (2) correlation with stemness/undifferentiated signatures defined by CytoTRACE (PMID: 31974247) and cancer stemness index (PMID: 29625051), etc. Most importantly, (3) tumor-initiating ability (self-renewal) is functionally defined by *in vitro* and *in vivo* limiting dilution assays, experiments that are now widely accepted in the field (PMID: 37537300). In regards to the comment regarding whether they are “tumor initiating/propagating cells or rather subsets of (resistant) cancer cells”, this would be very hard to delineate; however our past work has extensively shown that resistant cancer cells do exhibit elevated tumor-initiating potential. Above information is now incorporated in page 13 (Discussion) of our revised manuscript.

- The authors illustrated how EGFR silencing abrogates SPINK1-mediated self-renewal and chemoresistance. In view of EGFR implication on the resistance of HCC cells to RTK inhibitors like Lenvatinib, the authors should assess whether EGFR inhibitors used in the clinic recapitulates the effects observed on tumour plasticity shown by SPINK1 modulation and the relation with HCC resistance.

Reply: We thank the reviewer for this insightful suggestion to investigate clinically relevant EGFR inhibitors. We have now assessed the effect of the FDA-approved EGFR inhibitor (Erlotinib) in our models, as Erlotinib has been shown to inhibit stemness in hypoxic HCC cells (PMID: 34022398) and is the most commonly investigated EGFR inhibitor in HCC clinical trials (PMID: 16170173, 17623837, 23838576, 21953248, 19139433, 25547503).

Our results demonstrated that Erlotinib treatment (10 μ M) rescued the increases in chemoresistance, stem cell frequency, and activation of the EGFR-ERK-CDK4/6-E2F2 pathway induced by SPINK1

overexpression or rhSPINK1 treatment in MHCC97L cells. These findings provide valuable preclinical validation that targeting EGFR can recapitulate the effects of inhibiting the SPINK1 axis in our models. The data strengthen the clinical relevance of our proposed mechanism and support further evaluation of EGFR inhibition combined with SPINK1 blockade, as potential strategies to overcome resistance in HCC. This new data is now incorporated in **Supplementary Figure S5b-f** in our revised manuscript and below for easy reference.

Supplementary Figure S5

- The authors proposed a link of the EGFR-ERK-CDK4/6-E2F2 signalling circuit with self-renewal and chemoresistance. They should determine experimentally the contribution of this circuit in self-renewal and chemoresistance by targeting, for example, MEK/ERK and CDK4/6 with available inhibitors.

Reply: We thank the reviewer for encouraging us to directly validate the proposed EGFR-ERK-CDK4/6-E2F2 signaling mechanism. We chose the FDA-approved CDK4/6 inhibitor Palbociclib, which is approved for breast cancer treatment (PMID: 26324739). Our new finding, using Palbociclib (10ng/mL) in MHCC97L cells, showed that it is able to rescue the SPINK1-induced increases in self-renewal, chemoresistance, and activation of p-RB-E2F2 signaling. These new results provide strong evidence validating the functional role of this pathway in linking SPINK1 to the investigated phenotypes. This new data is now incorporated in **Supplementary Figure S6a-e** in our revised manuscript and below for easy reference.

Supplementary Figure S6

- In the HTVI experiments reported in Figure 9a-d, the authors performed a lentiviral injection with shSpINK1 (or control) after 2 weeks, followed by a 5-FU treatment after 3 weeks of HTVI. In the Methods sections, the authors claimed that in the NRasV12:Myr-AKT model “tumours started to grow at 2-3 weeks post-HTVI”, although no experimental data provide this assumption. Additional studies, ideally adding a luciferase reporter for longitudinal bioluminescence imaging, would contribute clarifying at which phase of tumour formation SPINK1 silencing and 5FU treatment act (before tumours are formed or when tumours are already formed). Moreover, SPINK1 silenced in shSPINK1 tumours should be documented by RT-qPCR and/or Western Blot analyses.

Reply: Regarding the NRasV12+Myr-AKT HCC model used in our study, we would like to emphasize that this is a well-established mouse model widely employed for investigating HCC. The utility and relevance of this model have been extensively documented in the literature (PMID: 21993994, 29885413). We apologize for not providing specific experimental data in our original manuscript to support the assertion that tumor development starts at 2-3 weeks post-HTVI in this HCC model. To address this concern, we conducted histological examination of mouse liver tissue at various time points (1, 2, 3 and 4 weeks post-

HTVI) using H&E staining. Our findings clearly demonstrate the presence of tumor nodules at 3 weeks post-HTVI, confirming the initiation of tumor growth within the specified timeframe. These additional images are shown below (tumor nodules indicated in white circle).

To address the reviewer’s suggestion for additional documentation of Spink1 silencing in shSpink1 tumors, please kindly note that we have already included both RT-qPCR and RNAScope staining data for Spink1 in our original manuscript. These analyses clearly demonstrated the successful knockdown of Spink1 in shSpink1 tumors. We apologize if this information was not evident in the original manuscript, and we have ensured that it is appropriately highlighted in the revised version. These data are in **Figure 9f-g** in our revised manuscript and below for easy reference.

Figure 9

- Transcriptomics data with list of differentially expressed genes in CD133+ vs CD133- cells must be available through open access platforms (e.g. EBI, ...).

Reply: We thank the reviewer for the valuable suggestion regarding the availability of our transcriptomics data comparing CD133+ and CD133- cells. We have now made our transcriptomics data openly accessible. The data is now deposited in the European Nucleotide Archive (ENA) under the accession code [PRJEB59278].

- Molecular weight (kDa) should be reported in all western blots shown in panels, e.g. on the right.

Reply: We have made the necessary revisions to address this concern. In our revised manuscript, we have included the molecular weight (kDa) markers in all our western blot images, including Figure 8b, Supplementary Figure S3d, S4a, S4g, S5a, S5f and S6e.

- The number of mice used for each study and for statistical analysis, should be reported in the corresponding figure legends.

Reply: We apologize for overlooking this matter. We have now included the number of mice used and experiments for each study and for statistical analysis in the corresponding figure legends.

Minor:

- Page 8 line 205: the following sentence should be corrected “We next explored delineating the receptor to which SPINK1 would bind on HCC cells to...”.

Reply: This sentence is now rephrased to read as: “Previous studies not in HCC have suggested EGFR to be a potential receptor of SPINK1 (24). Herein, we also explored if SPINK1 would act through EGFR to promote tumor-initiating and chemoresistance properties in HCC.”.

Reviewer #2 - HCC therapy, resistance, mouse models:

Thank you very much for inviting me to review the manuscript “SPINK1-induced tumor plasticity provides a therapeutic window for chemotherapy in hepatocellular carcinoma” by Man et al.

Building on their previous work, the authors investigated the role of CD133+ cells in hepatocellular carcinoma (HCC) and their response to chemotherapy. By comparing the transcriptomes of normal CD133+ liver cells and CD133+ HCC cells, they identified that SPINK1 expression was preferentially enriched in CD133+ HCC cells. Treatment with chemotherapy drugs, 5-FU and cisplatin, led to the upregulation of ELF3-mediated transcription and secretion of SPINK1, which further promoted tumor initiation, stemness, dedifferentiation, and chemoresistance in HCC cells. This effect was mediated through the binding of SPINK1 to EGFR, triggering the ERK-CDK4/6-E2F2 signaling cascade. Moreover, inhibiting SPINK1 function using neutralizing antibodies or lentivirus-mediated knockdown of Spink1 reduced HCC growth and improved chemotherapy response. Authors conclude that targeting SPINK1 could be a potential therapeutic strategy for HCC treatment.

This is a well written manuscript; the conclusions are supported by the data. Functional studies have been performed *in vitro* and *in vivo* to identify a novel mechanistic basis for chemoresistance. I have a few questions and comments:

Reply: We would like to thank the reviewer for the positive feedback and constructive feedback.

For figures, I would encourage the authors to individually point out what each figure depicts rather than use broad strokes. For instance, the entire Figure 1 is cited in the end of the first paragraph as Fig 1b-g. Instead it would be better to cite what fig 1b or 1c are depicting in the appropriate text.

Reply: The text is now rephrased to read as: “Importantly, where 5-FU treatment was administered as a standalone therapy (**Fig. 1b**), it did not demonstrate a reduction in tumor growth (**Fig. c-f**). In fact, it even resulted in an elevated stem cell frequency (**Fig. 1g**), which raises concerns about the potential for increased tumor recurrence if chemotherapy alone is utilized as a treatment strategy.”.

It is surprising that of the 615 genes upregulated in CD133+ mouse HCC cells, only 2 pass their screening of also being elevated in human HCC and predicting survival. Authors can maybe comment on why this might be the case.

Reply: We thank the reviewer for the question and observation regarding the limited overlap between the genes upregulated in CD133+ mouse HCC cells and those showing elevated expression in human HCC and predicting survival. We appreciate the opportunity to provide further clarification on this matter.

In our study, we employed three commonly used statistical methods, namely DESeq1, edgeR and linear models for microarray data (limma), to compare the gene expression profiles between non-tumor and tumor samples in the TCGA-LIHC cohort. Each method has its own statistical framework and underlying assumptions, leading to potentially different results. By applying these statistical methods, we aimed to identify genes that demonstrated robust and statistically significant upregulation in tumor samples, which may be indicative of their involvement in HCC development and progression. Concurrently with the identification of upregulated genes, we also performed survival analysis to explore the patient survival outcomes. Specifically, we selected genes that demonstrated significant upregulation in tumor samples with poor survival.

Considering the stringency of our screening process and the use of multiple statistical methods, it is not surprising that only two genes, SPINK1 and B4GALNT1, passed the selection criteria among the initial set of 615 upregulated genes. These two genes demonstrated elevated expression in both CD133+ mouse HCC cells and human HCC samples and their expression levels were found to be associated with patient survival. We believe that this stringent approach, coupled with the use of multiple statistical methods, helps to identify robust and clinically relevant candidates. Therefore, the small number of genes passing the screening among the initial 615 upregulated genes is not unexpected.

In Figure 2i the CD133+ cells are not visualized well. Moreover, in Figure 2i, Spink1 appears to be overexpressed diffusely across the tumor and is not restricted to the Prom1+ cells. However, in Figure 2b, authors show that Spink1 mRNA expression is mostly restricted to the Prom1+ cells, can they explain this?

Reply: We appreciate the reviewer’s observation regarding the visualization of CD133+ cells in Figure 2i and the apparent diffuse overexpression of Spink1 across the tumor in that figure. We would like to address this and provide an explanation for this discrepancy by considering that HCC tumors may not be the sole source of Spink1 expression.

Prior to sorting Prom1+/- cells, we selectively picked tumor nodules from the mouse HCC liver. Furthermore, the sorting of Prom1+/- cells was gated based on double negativity for CD45 and TER119 to

exclude immune cells from the tumor bulk. Therefore, the results presented in Figure 2b solely reflect Spink1 mRNA expression in HCC tumor cells and no other cell types.

To investigate the expression of SPINK1 in different cell types within the HCC tumor microenvironment (TME), we analysed a publicly available single-cell RNA sequencing dataset (GSE151530). This analysis revealed that while the majority of SPINK1 is expressed in tumor cells, there is also some expression in immune cells and stromal cells. In order to validate these findings, we performed co-detection experiments using RNAScope to detect Spink1 mRNA and multiplex immunohistochemistry (IHC) to identify immune and stromal cell markers in mouse HCC liver tissues. Our results demonstrated that Spink1 is expressed in immune cells (CD45+ and CD3+ cells) and fibroblast (α -SMA+ cells), with minimal expression in endothelial cells (CD31+ cells). Consequently, we speculate that the staining of Spink1 in the mouse HCC liver tissue (Figure 3i) may be attributed, at least in part to these other cell types rather than exclusively to tumor cells. These new data is now incorporated in **Supplementary Figure S8a-g** in our revised manuscript and below for easy reference.

Supplementary Figure S8a-g

Several prior papers have pointed out that SPINK1 is overexpressed in HCC and is associated with poor prognosis (PMID: 23527199, PMID: 34595116, PMID: 32066292, PMID: 27028242) so a lot of the correlative results shown in figure 3 can potentially be moved to the supplementary data so we can focus on the functional analysis presented in later figures.

Reply: Thank you for the comment. Past literature has only reported on SPINK1 overexpression in HCC, which matches with our data shown in Figure 3e and 3i. Other parts of our Figure 3 consist important correlative data showing SPINK1 to be upregulated at a more undifferentiated/stemness state and to be enriched with more stemness signatures which has not been reported elsewhere. We feel that this information should be retained in the main figure.

One of the key results is that “Knockdown of SPINK1 also resulted in a profound decrease in the ability of cells to resist 5-FU and cisplatin, as indicated by enhanced rates of apoptosis in the knockout clones as compared to the control (Fig 4h)”. Even though the quantitative graphs below show a significant change the representative flow images show only a 4-5% change with Spink1 knockdown. Can authors comment?

Reply: We thank the reviewer for bringing this to our attention. This particular Annexin V-PI apoptosis assay was repeated 4 times and the quantitative graphs do show a significant change, though as one may expect each experiment may yield a little bit of a varying result, though in our case all following the same correct trend. We have now replaced the representative flow images in Figure 4h that we feel would be a more accurate visualization of the enhanced rate of apoptosis observed in the knockdown clones compared to the control. Below revised image is incorporated for easy reference.

Figure 4

Moreover, it's not clear if Fig 4h and 4g are from *in vivo* or *in vitro* data.

Reply: We apologize for the lack of clarify in Figure 4h and 4g regarding whether they represent *in vivo* or *in vitro* data. We have now revised the figures in our manuscript to provide clear labels indicating whether the data is derived form *in vitro*, *in vivo* or *ex vivo* experiments. The updated and labelled images can be found in Figure 4b, 4c and 4g of our revised manuscript and below for easy reference.

Figure 4

In the discussion authors say “we report on the role of endogenous/secretory SPINK1 in inducing HCC to a more stem/progenitor state and sensitizing HCC cells to chemotherapy.” I would say making HCC cells resistant to chemotherapy or desensitizing cells to chemotherapy.

Reply: Revised sentence now reads: “Here, we report on the role of endogenous/secretory SPINK1 in inducing HCC to a more stem/progenitor state and making HCC cells resistant to chemotherapy.”. Thank you for the suggestion.

Authors should discuss how they expect SPINK1 inhibitors to be used in HCC. Do they think they should be used as adjuvant therapy with TACE? Or in patients who don’t respond to TACE?

Reply: We appreciate the reviewer’s insightful comment regarding the potential use of SPINK1 inhibitors in the treatment of HCC. In response to the question, we believe that SPINK1 inhibitors could be utilized as an adjuvant therapy in combination with TACE or in patients who do not respond to TACE.

Several studies have shown that combining TACE with tyrosine kinase inhibitors (TKIs) and/or immune checkpoint inhibitors, either as post- or pre-treatment, can effectively prolong the survival of HCC patients (PMID: 33521054, PMID: 31801872). Additionally, the phase III LAUNCH clinical trial demonstrated improved clinical outcomes in HCC patients treated with a combination of Lenvatinib and TACE (PMID: 35921605). Our study has revealed that the combination treatment of Spink1 knockdown and chemotherapy can reduce tumor growth and prolong survival in a HCC mouse model. Based on these findings, we propose that combining TACE with a SPINK1 inhibitor could be a potential therapeutic approach for HCC patients. This combination therapy may enhance the efficacy of TACE and improve treatment outcomes. Furthermore, we have observed that chemotherapy treatment alone can lead to the enrichment of the CD133+ liver cancer stem cell subset and an increase in SPINK1 expression. This may explain why some HCC patients exhibit resistance to chemotherapy. Therefore, HCC patients who are resistant to TACE and exhibit high serum SPINK1 levels could potentially benefit from treatment with a SPINK1 inhibitor. Such treatment could sensitize HCC cells to TACE and improve the overall treatment response. However, it is important to note that further investigation is required to determine the optimal timing and sequencing of SPINK1 inhibitor administration in relation to TACE. Clinical studies and preclinical models are needed to assess the effectiveness and safety of combining SPINK1 inhibitors with TACE in HCC patients. These studies will provide valuable insights into the potential therapeutic strategies for improving the outcomes of HCC treatment.

The above information has now been incorporated in the revised Discussion on page 15.

The same authors have previously suggested that “CD133+ HCC cells contribute to chemoresistance preferential activation of Akt/PKB and Bcl-2 cell survival response.” Authors should discuss if these two mechanisms are complementary or unconnected.

Reply: We appreciate the reviewer’s comment regarding our previous studies which suggested that CD133+ HCC cells contribute to chemoresistance through the preferential activation of Akt/PKB and the Bcl-2 cell survival response (PMID: 22617155). The reviewer raised an important question about the relationship between the SPINK1 signaling pathway and the AKT/PKB and the Bcl-2 pathway, and whether they are complementary or unconnected.

Previous works revealed that the preferential activation of Akt/PKB and upregulation of Bcl-2 in CD133+ HCC cells promoted cell survival and increase chemoresistance (PMID: 22617155). This pathway serves as a protective mechanism, enabling CD133+ cells to evade cell death induced by chemotherapy agents. The Akt/PKB and the Bcl-2 pathway likely plays a significant role in chemoresistance and cell proliferation through survival signaling pathway. Conversely, our findings showed that SPINK1 overexpression in HCC cells does not significantly impact cell proliferation or cell cycle progression. Instead, our results demonstrated that SPINK1 primarily influences tumor plasticity to induce chemoresistance in HCC. This suggests that SPINK1 signaling pathway may exert its effect through alternative mechanisms that are distinct from the promotion of cell proliferation in Akt/PKB and the Bcl-2 cell survival response. While the specific relationship between the SPINK1 signaling pathway and the AKT/PKB and the Bcl-2 pathways remains to be fully elucidated, our findings suggest that these pathways may act independently in mediating different aspects of chemoresistance in HCC cells. The SPINK1 pathway, rather than affecting cell proliferation, may primarily influence tumor plasticity, leading to enhanced resistance to chemotherapy. In contrast, the Akt/PKB and the Bcl-2 pathways likely contribute to the survival and protection of CD133+ cells from chemotherapy-induced cell death. However, it is important to note that further investigation is required to fully elucidate the specific interactions and crosstalk between these pathways in CD133+ HCC cells. Additional studies may provide insights into the extent of their connection and whether they act independently or cooperatively to confer chemoresistance. Moreover, exploring therapeutic strategies targeting these pathways simultaneously may hold promise for overcoming chemoresistance in HCC.

Above information is now incorporated on page 16 (Discussion) of our revised manuscript.

Reviewer #3 - HCC, RNA-seq:

The manuscript entitled “SPINK1-induced tumor plasticity provides a therapeutic window for chemotherapy in hepatocellular carcinoma” submitted by Man et al made attempted to show SPINK1 closely associated with CD133+ HCC, liver development, and tumor dedifferentiation. They found that enhanced transcriptional activity of SPINK1 was mediated by promoter binding of ELF3, which like CD133, increased following chemotherapy treatment. Functionally, SPINK1 overexpression enhanced tumor initiation, self-renewal, and chemoresistance, with a deregulated EGFR-ERK-CDK4/6-E2F2 signaling engaged to induce dedifferentiation of HCC cells into their ancestral lineages. They eventually proposed that targeting oncofetal SPINK1 may represent a good therapeutic option for HCC clinics. Although the authors presented several lines of interesting data to make the final conclusions, the whole study suffers from the lacking of appropriate elucidation of functional mechanisms, with the majority of the

experimental data basically not supporting the hypothesis. Significant improvements are essential for the current work.

Reply: We thank the reviewer for reviewing our manuscript. We appreciate your valuable feedback and acknowledge your concerns regarding the clarification of functional mechanisms and the support for our hypothesis. We apologize for any confusion caused by the presentation of our data and the limited explanation of underlying mechanisms. We hope this revised version of the manuscript has addressed your concerns.

Some minors:

1. Fig. 1a and Supplementary Fig. S1a intended to show that 5-FU or cisplatin enriched for a CD133/Prom1+ liver tumor initiating/propagating cell subset. However, the flow cytometry analysis for Prom1 expression in mouse NRasV12+Myr-AKT HTVI HCC tumors with DMSO or 5-FU treatment was demonstrated, the reason using these mice (hydrodynamic tail vein injected (HTVI) NRasV12+Myr-AKT proto-oncogenes-driven HCC tumor mouse model, rather than a de novo tumorigenesis model) remains fairly unclear, should be explained in the text.

Reply: We appreciate the reviewer's question regarding the choice of the NRasV12+Myr-AKT HTVI HCC mouse model in our study. The rationale for using this specific mouse model is based on the frequency of activation of both the RAS/MAPK and PI3K/AKT/mTOR pathways in human HCC. It has been reported that approximately 50% of HCC patients exhibit activation of these pathways, making it a clinically relevant model for studying HCC biology and therapeutic interventions (PMID: 26054909). The NRasV12+Myr-AKT proto-oncogenes-driven HCC tumor mouse model is a well-established and widely used model for studying HCC. This model efficiently induces liver tumors in a relatively short time, allowing for the investigation of various aspects of HCC biology and therapeutic interventions in a timely manner (PMID: 21993994). We apologize for not providing a clear explanation in the manuscript regarding the choice of this mouse model. In our revised manuscript on page 5, we have included a statement in the text to clarify the rationale for using this mouse model and its relevance to human HCC.

2. Page 5, the 1st paragraph. Notably, 5-FU treatment alone did not reduce tumor growth and importantly also led to an elevated stem cell frequency suggesting a higher chance of tumor recurrence if only chemotherapy is utilized (Fig. 1b-g). The sentence appears to be quite embarrassed and needs to be rewritten.

Reply: Revised sentence now reads: "Importantly, where 5-FU treatment was administered as a standalone therapy (**Fig. 1b**), it did not demonstrate a reduction in tumor growth (**Fig. c-f**). In fact, it even resulted in an elevated stem cell frequency (**Fig. 1g**), which raises concerns about the potential for increased tumor recurrence if chemotherapy alone is utilized as a treatment strategy."

3. Page 5, the 2nd paragraph. While we and others have now demonstrated a functional role of CD133/Prom1 in HCC, CD133 unfortunately is not specific to HCC and is also expressed in the regenerating liver. Poorly organized.

Reply: Revised sentence now reads: "While CD133/Prom1 has been demonstrated to have a functional role in HCC by us and others, it is important to note that CD133 is not specific to HCC and is also expressed in the regenerating liver."

4. Page 5. Upon further correlation with survival using human clinical samples extracted from TCGA-LIHC, IGALS, SPINK1 and B4GALNT1 were identified as potential candidates (Fig. 2a and Supplementary Fig.

S2a, b). The selection criteria were not provided at all. Furthermore, supplementary Fig. S2a, b are too busy and should be simplified.

Reply: In our revised manuscript, we have included the selection criteria for identifying potential candidates from the mouse models and human clinical samples from TCGA-LIHC cohort. Furthermore, we have taken the reviewer’s feedback and have simplified and reorganized Supplementary Figure S2a and S2b to enhance readability. We have made efforts to reduce visual complexity while ensuring that the relevant information is still conveyed effectively. The updated figures can be found in **Figure 2a** and **Supplementary Figure S2a and S2b** of our revised manuscript and below for easy reference.

Figure 2

Supplementary Figure S2

5. Page 6. SPINK1 is not expressed in the normal liver nor many other major organs and immune cells, making it an attractive therapeutic target. Did the authors analyse SPINK1 expression pattern in the overall liver tumor microenvironment, rather than simply considering the tumor bulk or CD133+ cells? Chemotherapy induces SPINK1 expression in the tumor stroma, although not tumor cells, how to reconcile the differential responses?

Reply: We thank the reviewer for the suggestion regarding the analysis of SPINK1 expression in the liver tumor microenvironment (TME), particularly after chemotherapy treatment. We have now conducted an analysis using single-cell RNA sequencing of a publicly available dataset (GSE151530). The analysis revealed that SPINK1 is primarily expressed in malignant cells, with minimal expression in immune cells and stromal cells. These findings suggest that SPINK1 is not only expressed in liver tumor cells but also in immune cells and stromal cells within the TME.

In previous study on prostate cancer, it has been shown that chemotherapy-treated stromal cells secrete SPINK1, promoting cancer progression (PMID: 30333494). To investigate whether chemotherapy treatment in HCC cells increases SPINK1 expression in cell types other than tumor cells, as demonstrated in our study, we performed co-detection of Spink1 mRNA using RNAScope and protein expressions of immune cell markers (CD45 and CD3) and stromal cell markers (α -SMA and CD31) using multiplex immunohistochemistry (IHC) in mouse HCC liver tissues treated with 5-FU or cisplatin. Our findings revealed that while immune cells and stromal cells do express SPINK1, only fibroblast cells (α -SMA+) showed an increase in SPINK1 expression following 5-FU and cisplatin treatment in HCC. However, further investigation is required to determine the exact functional role of fibroblast-secreted SPINK1 in HCC.

These new data is now incorporated in **Supplementary Figure S8** in our revised manuscript and below for easy reference.

Supplementary Figure S8

6. Supplementary Figure S1b shows representative image of livers from NRasV12+Myr-AKT HTVI HCC mice treated with DMSO, 5-FU or cisplatin with Spink1 mRNA staining by RNAScope. However, the data need to be accompanied by protein level examinations such as immunohistochemistry staining against Spink1, to allow the conclusions make further sense.

Reply: We appreciate the reviewer's comment regarding the protein level examination of Spink1 in our mouse models. We acknowledge the importance of validating mRNA expression with protein level analysis for a comprehensive understanding of Spink1 expression. Unfortunately, despite our efforts, we were unable to find a commercially available anti-mouse Spink1 antibody suitable for immunohistochemistry (IHC) staining. We attempted to perform IHC using multiple anti-human SPINK1 and anti-mouse Spink1 antibodies; however, the results were not satisfactory. Therefore, as an alternative approach, we utilized RNAScope to stain Spink1 mRNA expression in mouse HCC liver tissues. While RNAScope provides spatial visualization of Spink1 expression, we understand that protein level examination would further enhance the interpretation of the results.

To support the correlation between SPINK1 mRNA and protein levels in our study, we have included additional data showing a positive correlation between SPINK1 mRNA and protein levels, as well as the levels of secreted SPINK1, in human HCC cell lines. These findings suggest that there is a potential positive correlation between Spink1 mRNA and protein expression in our mouse model. The figures below illustrate a positive correlation between the expression of SPINK1 at the mRNA level, as determined by RT-qPCR, and the protein level, as well as the secreted form of SPINK1, both measured by ELISA.

Nonetheless, we realize that this is just a correlation. We hope that the reviewer can understand our limitation.

7. Page 6. Spink1 was also found to be enriched in HCC tumors enriched for a CD133+ liver tumor initiating/propagating cell subset following either 5-FU or cisplatin treatment, why? Did the authors investigate the mechanism(s) underlying the induction or enrichment of Spink1 expression in HCC tumors? How about the stromal cells, such as fibroblasts and immune cell subtypes?

Reply: The reviewer may have overlooked. In fact, our findings suggest that chemotherapy treatment in HCC cells leads to an increase in the expression of CD133 and ELF3. We have observed that ELF3 binds to the upstream transcriptional start site of SPINK1, promoting its transcriptional activity. Above data is presented in our original Figure 7. Notably, ELF3 is also preferentially expressed in Prom1+ cells, as shown in our original Supplementary Figure S4c. To further investigate the relationship between ELF3 and CD133, we examined the expression of CD133 in HCC cells with ELF3 knockout. Interestingly, we did not observe any changes in the expression of CD133/Prom1 at the protein level, indicating that ELF3 does not directly regulate the expression of CD133/Prom1 (Supplementary Figure S4g; incorporated below for easy reference). However, it remains to be determined whether CD133 regulates the expression of ELF3.

Further investigation is needed to examine the expression of ELF3 in CD133 knockdown HCC cells to shed light on this potential regulatory relationship.

Supplementary Figure S4

8. Just according to the data that SPINK1 associates closely with CD133, it is not appropriate to infer that SPINK1 might be associated with liver development (Supplementary Fig. S3c). The speculation is not valid.

Reply: The sentence now reads as “We next explored the involvement of SPINK1 in liver development.”.

9. Page 7. Using CytoTRACE, a computational framework for predicting differentiation from former analysis of scRNA-seq data, the authors stated to have found high SPINK1 expression overlapping with the less differentiated hepatic cells (Fig. 3d). The opposite was indicated in fact.

Reply: The red color in the CytoTRACE plot (Figure 3d, top) represents less differentiated cells, while the purple color represents more differentiated cells. The SPINK1 expression plot (Figure 3d, bottom) shows higher expression of SPINK1 in less differentiated cells and lower expression levels in more differentiated cells, suggesting high SPINK1 expressing cells are overlapped with less differentiated cells. Figure 3d is included below for easy reference. Perhaps the reviewer may have made a mistake in their interpretation.

Figure 3

10. It was explained that patients in TCGA-LIHC cohort were ranked according to the transcriptomic level of SPINK1 in their tumors. However, 10% of highest-ranking patients were marked as high SPINK1, while 10% of lowest-ranking patients were marked as low SPINK1, where was this indicated in Fig. 3f-g? Should be clearly marked through.

Reply: We thank the reviewer for bringing up the need for clearer labelling in Figure 3f regarding the ranking of patients based on SPINK1 transcriptomic levels. We appreciate the reviewer’s feedback, and we have made the necessary revisions to address this concern.

In the revised version of Figure 3f, we have added labels indicating “High SPINK1” and “Low SPINK1” to clearly indicate the ranking categories. This modification ensures better visual clarity and helps in understanding the patient categorization based on SPINK1 expression levels. The revised **Figure 3f** is now incorporated in our revised manuscript and below for easy reference.

Figure 3

11. Fig. 3i. High proteomic SPINK1 levels significantly correlated with a worst overall survival. Kaplan-Meier analysis of the overall survival in HCC patients with high and low SPINK1 scoring in their tumor samples. * $p < 0.05$ by log-rank test. SPINK1 expression either low or high, with the lines corresponding to each status basically missing (had better follow the presentation manner of Fig. 8a).

Reply: We apologize for any confusion caused by the lack of clear labelling of Figure 3i. In the revised version of Figure 3i, we have added labels indicating “SPINK1 Low” and “SPINK1 High” to clearly indicate the status of SPINK1 expression.

Figure 3

12. Fig. 3j shows the transcriptomic level of SPINK1 in tumors (T) and paired adjacent non-tumoral livers (NT) of NASH related HCCs. How many patients’ data were acquired and assessed? Should be well stated.

Reply: We apologize for overlooking this matter. In the revised version of Figure 3j, we have included the information that the data presented corresponds to a total of 15 patients.

Figure 3

13. Fig. 6a. Co-immunoprecipitation analysis for validation of EGFR as an interacting protein partner of SPINK1 in Huh7 cells. The data are incomplete, as input samples need to be fully provided to cover all the 4 lanes of lysates, rather than simply showing the input of SPINK1, EGFR and actin in only one lane.

Reply: We thank the reviewer for bringing up this issue regarding the incompleteness of the co-immunoprecipitation analysis in Figure 6a. In the revised Figure 6a, we have included the input samples for all four lanes of lysates, providing a more comprehensive view of the experimental setup. The revised Figure 6a is now incorporated in our revised manuscript and below for easy reference.

Figure 6

14. Fig. 6c, d show data from *in vitro* limiting dilution analysis for frequency of TICs (c) MHCC97L with rhSPINK1 treatment or (d) SPINK1 overexpression in the presence or absence of shRNA against EGFR stably transduced. The authors had better display the growth curves or proliferative potential of HCC cells to allow a more straightforward comparison.

Reply: Thank you for the comment but we are not too sure what the reviewer is requesting for here. Figure 6c and 6d shows results for self-renewal ability and not proliferation.

15. Supplementary Fig. S4a indicates the western blot data of SPINK1 regulation through an EGFR axis, with SPINK1 overexpressed in MHCC97L cells with EGFR knockdown. The quality of WB data are generally poor, need to be redone.

Reply: We apologize for the poor quality of the image. We have now redone the western blot experiments, including the expression of SPINK1, to ensure improved quality of the data. Additionally, we have also taken the opportunity to enhance the quality of all other western blot images presented in the manuscript.

The revised **Supplementary Figure S4a** is now incorporated in our revised manuscript and below for easy reference.

Supplementary Figure S4

16. Page 9. To seek hints into the possible upstream regulatory element that leads to SPINK1 overexpression in HCC, gene transcription factors binding to SPINK1 promoter was predicted from a public collection of ChIP-seq data website - Gene Transcription Regulation Database (GTRD). Grammar problems throughout the sentence.

Reply: The revised sentence now reads as: “In order to investigate the potential upstream regulatory element responsible for SPINK1 overexpression in HCC, we utilized the Gene Transcription Regulation Database (GTRD), a publicly available collection of ChIP-seq data, to predict gene transcription factors binding to the SPINK1 promoter. Our analysis revealed that the cell cycle-related transcription factor ELF3 contains two high-scoring predicted binding sites, including GAAAAGGAAAAA at -3860 to -3848 and AAGGAAGAAATAA at -2047 to -2035 (Fig. 7a).”.

17. Supplementary Fig. S4g, lacks the expression data of SPINK1.

Reply: We apologize for overlooking this. In this revised version of the manuscript, we have included the expression data of SPINK1 in the western blot. The revised **Supplementary Figure S4g** is now incorporated in our revised manuscript and below for easy reference.

Supplementary Figure 4

18. Page 10. As E2F is widely known to be linked to cell cycle regulation, we first examined if SPINK1 may promote HCC through altering cell cycle proliferation. Sick sentence, needs to be rewritten.

Reply: The revised sentence now reads as: “Given the well-established association between E2F and cell

cycle regulation, our initial investigation focused on determining if SPINK1 promotes HCC by influencing cell cycle progression.”.

19. Fig. 8b. Western blot shows the expression of SPINK1, phosphorylated and total MEK, 958 phosphorylated and total ERK, CDK4, CDK6, Cyclin D1, phosphorylated and total Rb, E2F2, etc. The levels of PCNA and MCM3 need to be examined, with Cyclin A2 to be analysed in parallel.

Reply: We appreciate the reviewer’s suggestion to examine the expression levels of PCNA, MCM3 and Cyclin A2, as they are indeed important factors in cell proliferation and cell cycle progression. We have now performed western blot analysis to investigate their expression in HCC cell lines where SPINK1 expression is modulated. Using specific antibodies targeting MCM3 (1:1000, 4003, Cell Signaling Technology), PCNA (1:1000, ab29, Abcam), Cyclin A2 (1:1000, 91500, Cell Signaling Technology), we evaluate their expression in SPINK1 manipulated HCC cells by western blot. The results indicated that the expression levels of PCNA, MCM3 or Cyclin A2 were not altered upon SPINK1 manipulation. These findings are consistent with the observations presented in Figure 8a, Supplementary S6f and S7, further supporting the conclusion that SPINK1 does not significantly affect cell proliferation or cell cycle progression. We have incorporated this new data into all the relevant western blot figures in our revised manuscript. Below is an example of one of the western blot figures, Figure 8b, which includes the expression data for SPINK1, phosphorylated and total MEK, phosphorylated and total ERK, CDK4, CDK6, Cyclin D1, phosphorylated and total RB, E2F2, MCM3, PCNA and Cyclin A2 for reference.

Figure 8

20. Fig. 8d shows GSEA data on HCC patients segregated into high/low SPINK1 expression using Reactome G1/S Transition, WP G1 to S Cell Cycle Control, and proliferative gene sets without stemness genes signatures in TCGA-LIHC cohort. The GSEA graphs are too small and had better be enlarged to allow visualization.

Reply: We have now enlarged the GSEA graphs in Figure 8d to improve visibility and ensure that the results can be properly visualized and interpreted by the readers.

21. The manuscript demonstrated that 5-FU or cisplatin chemotherapy would functionally enrich for CD133/Prom1, as a side effect of treatments, suggesting that while CD133 may be proliferative, they may

also be driven by other mechanisms. SPINK1-mediated tumor plasticity is only one of such potential mechanisms, the authors need to integrate into discussion.

Reply: We agree that it is essential to consider other potential mechanisms that may contribute to the enrichment of CD133+ cells in response to chemotherapy. We thank the reviewer for highlighting the importance of integrating these potential mechanisms into the discussion.

The functional enrichment of CD133/Prom1 following 5-FU or cisplatin treatment, as observed in previous and our present studies, suggests that many different mechanisms are likely contributing to its enrichment. In addition to SPINK1-mediated tumor plasticity, the activation of other signaling pathways could drive the functional enrichment of CD133+ cells in response to chemotherapy. Previous studies have reported the activation of the NOTCH pathway in CD133+ HCC cells following treatment with 5-FU. Inhibition of the NOTCH pathway has been shown to sensitize CD133+ HCC cells to chemotherapy by inducing apoptosis through the activation of BCL-2-binding component 3 (BBC3)-mediated apoptosis (PMID: 32173531). In our previous investigations, we observed the preferential activation of Akt/PKB and upregulation of Bcl-2 in CD133+ HCC cells, which contributed to their resistance to 5-FU treatment (PMID: 17891174). It is conceivable that these signaling pathways play a role in the functional enrichment of CD133+ cells following chemotherapy. Moreover, the tumor microenvironment is known to influence CD133 expression and enrichment. The microenvironment provides supportive signals and cues that favour the survival and expansion of CD133+ cells. In our previous study, we demonstrated that chemotherapy treatment enriched thrombospondin 2 (THBS2)-deficient CD133+ liver CSCs and promoted HCC progression. This enrichment is facilitated by matrix softness-induced histone H3 modifications (PMID: 33717837). Further studies exploring the interplay between CD133+ cells and the tumor microenvironment will help elucidate the impact of these factors on CD133 expression and the mechanisms driving its enrichment following chemotherapy. Our present study focused on SPINK1-mediated tumor plasticity as one potential mechanism contributing to the functional enrichment of CD133+ cells following chemotherapy. However, we acknowledge that additional mechanisms, including survival signaling pathways and the tumor microenvironment, may also play significant roles.

Above information is now incorporated on page 15 (Discussion) of our revised manuscript.

Reviewers' Comments:

Reviewer #1:

Remarks to the Author:

The authors have addressed most of my concerns. The level of novelty in relation to previous publication remains an issue, which should be managed as editorial decision.

Reviewer #2:

Remarks to the Author:

Thank you very much for inviting me to review the revised version of the manuscript "SPINK1-induced tumor plasticity provides a therapeutic window for chemotherapy in hepatocellular carcinoma" by Man et al.

Building on their previous work, the authors investigated the role of CD133+ cells in hepatocellular carcinoma (HCC) and their response to chemotherapy. By comparing the transcriptomes of normal CD133+ liver cells and CD133+ HCC cells, they identified that SPINK1 expression was preferentially enriched in CD133+ HCC cells. Treatment with chemotherapy drugs, 5-FU and cisplatin, led to the upregulation of ELF3-mediated transcription and secretion of SPINK1, which further promoted tumor initiation, stemness, dedifferentiation, and chemoresistance in HCC cells. This effect was mediated through the binding of SPINK1 to EGFR, triggering the ERK-CDK4/6-E2F2 signaling cascade. Moreover, inhibiting SPINK1 function using neutralizing antibodies or lentivirus-mediated knockdown of Spink1 reduced HCC growth and improved chemotherapy response. Authors conclude that targeting SPINK1 could be a potential therapeutic strategy for HCC treatment.

Authors have adequately answered my questions and also those of the other two reviewers, and made revisions which has improved the manuscript. I have no further questions.

The abstract doesn't fully capture the results and is a little vague. Would suggest they edit the abstract to include the following points-

They highlight the role of SPINK1 in liver development in the abstract. However, the only data that supports this in the manuscript is correlation of SPINK1 expression with hepatoblast and fetal hepatocyte (FH) cell clusters. No functional embryonic knockdown experiments were presented, would avoid highlighting this point in the abstract and focus on HCC, on which most of the data is presented.

They need to mention that the results were specifically related to 5-FU and cisplatin and not broadly use the term chemotherapy.

This is an important result in the paper but does not find place in the abstract, would recommend including-

"Depleting SPINK1 function by neutralizing antibody treatment or in vivo lentivirus-mediated Spink1 knockdown dampened HCC cancer growth and their ability to resist chemotherapy."

Reviewer #3:

Remarks to the Author:

The revised manuscript largely support the authors' final conclusions and claims, although additional evidence is highly desired to consolidate their new data. Enough details should be provided in the methods for the work to be reproduced by other colleagues in cancer research. The authors tried to demonstrate that chemotherapy upregulates SPINK1 via ELF3, where SPINK1 would subsequently enhance self-renewal, dedifferentiation and the cancer stem cell phenotype via activation of EGFR-ERK-CDK4/6-Cyclin D1-E2F2 signaling pathway, conferring resistance to treatment. Although the authors claimed that their findings may advance understanding of the molecular mechanisms through which SPINK1 modulates therapeutic response in HCC, the novelty

needs to be further addressed in the text, as also suggested by other reviewers.

SPINK1-induced tumor plasticity provides a therapeutic window for chemotherapy in hepatocellular carcinoma

NCOMMS-23-16086A

REVIEWERS' COMMENTS

Reviewer #1 (Remarks to the Author):

The authors have addressed most of my concerns. The level of novelty in relation to previous publication remains an issue, which should be managed as editorial decision.

Reply: We thank the reviewer's feedback regarding the concern of novelty in relation to previous publication. We acknowledge that certain aspects of our results have been reported in the context of HCC and other cancer types. However, we would like to emphasize that our study does present several novel findings that contribute to the existing knowledge in the field.

Firstly, while the binding of SPINK1 to EGFR has been demonstrated in pancreatic cancer (PMID:24619958), our study is the first to show this interaction in HCC cells. This expands the understanding of the molecular interactions involving SPINK1 beyond its established role in pancreatic cancer.

Secondly, although previous studies have linked SPINK1 overexpression in HCC with poor prognosis (PMID: 34569662), our study reveals a significant association between SPINK1 expression and the histopathological grade of HCC. This finding further suggests the relationship between SPINK1 and less differentiated HCC.

Thirdly, while the impact of SPINK1 overexpression in chemosensitivity in HCC has been investigated before (PMID: 35924241), our study delves deeper into the underlying mechanisms by demonstrating how SPINK1 specifically promotes chemoresistance in HCC through the induction of tumor plasticity. This connection between SPINK1 expression and tumor plasticity processes that underlie chemoresistance in HCC is a novel contribution, shedding light on the complex molecular mechanisms involved in therapeutic response.

Last but not least, our study is the first to investigate the proof-of-principle therapeutic experiment of Spink1 knockdown in an HCC mouse model. We observed that Spink1 knockdown effectively reduced tumor growth and overcame chemoresistance in vivo. This finding highlights the potential of targeting SPINK1 as a therapeutic approach for chemoresistant HCC patients.

Overall, we believe that our study significantly advances the understanding of the molecular mechanisms through which SPINK1 modulates therapeutic response in HCC. While certain aspects of our findings have been reported in different contexts, the specific combinations of novel findings presented in our study make a unique and valuable contribution to the field. We hope that this clarification addresses the concern of novelty and reinforces the significance of our research.

Reviewer #2 (Remarks to the Author):

Thank you very much for inviting me to review the revised version of the manuscript "SPINK1-

induced tumor plasticity provides a therapeutic window for chemotherapy in hepatocellular carcinoma” by Man et al.

Building on their previous work, the authors investigated the role of CD133+ cells in hepatocellular carcinoma (HCC) and their response to chemotherapy. By comparing the transcriptomes of normal CD133+ liver cells and CD133+ HCC cells, they identified that SPINK1 expression was preferentially enriched in CD133+ HCC cells. Treatment with chemotherapy drugs, 5-FU and cisplatin, led to the upregulation of ELF3-mediated transcription and secretion of SPINK1, which further promoted tumor initiation, stemness, dedifferentiation, and chemoresistance in HCC cells. This effect was mediated through the binding of SPINK1 to EGFR, triggering the ERK-CDK4/6-E2F2 signaling cascade. Moreover, inhibiting SPINK1 function using neutralizing antibodies or lentivirus-mediated knockdown of Spink1 reduced HCC growth and improved chemotherapy response. Authors conclude that targeting SPINK1 could be a potential therapeutic strategy for HCC treatment.

Authors have adequately answered my questions and also those of the other two reviewers, and made revisions which has improved the manuscript. I have no further questions.

The abstract doesn't fully capture the results and is a little vague. Would suggest they edit the abstract to include the following points-

They highlight the role of SPINK1 in liver development in the abstract. However, the only data that supports this in the manuscript is correlation of SPINK1 expression with hepatoblast and fetal hepatocyte (FH) cell clusters. No functional embryonic knockdown experiments were presented, would avoid highlighting this point in the abstract and focus on HCC, on which most of the data is presented.

They need to mention that the results were specifically related to 5-FU and cisplatin and not broadly use the term chemotherapy.

This is an important result in the paper but does not find place in the abstract, would recommend including-

“Depleting SPINK1 function by neutralizing antibody treatment or in vivo lentivirus-mediated Spink1 knockdown dampened HCC cancer growth and their ability to resist chemotherapy.”

Reply: Thank you for the reviewer’s suggestion. We have revised the abstract with (1) the deletion of the point related to liver development, (2) replacement of “chemotherapy” to “5-FU and cisplatin treatment” and (3) included in the abstract the suggested statement “Depleting SPINK1 function by neutralizing antibody treatment or in vivo lentivirus-mediated Spink1 knockdown dampened HCC cancer growth and their ability to resist chemotherapy.”. We feel that these changes have now improved the abstract by accurately reflecting the scope and findings of our study. Revised abstract is in page 2 of our revised manuscript.

Reviewer #3 (Remarks to the Author):

The revised manuscript largely support the authors' final conclusions and claims, although additional evidence is highly desired to consolidate their new data. Enough details should be provided in the methods for the work to be reproduced by other colleagues in cancer research. The authors tried to demonstrate that chemotherapy upregulates SPINK1 via ELF3, where SPINK1 would subsequently enhance self-renewal, dedifferentiation and the cancer stem cell phenotype

via activation of EGFR-ERK-CDK4/6-Cyclin D1-E2F2 signaling pathway, conferring resistance to treatment. Although the authors claimed that their findings may advance understanding of the molecular mechanisms through which SPINK1 modulates therapeutic response in HCC, the novelty needs to be further addressed in the text, as also suggested by other reviewers.

Reply: We appreciate the reviewer's feedback. Regarding the concern about providing sufficient methodological details, we have made sure to include additional information in the Methods section of the revised manuscript to enable other colleagues in cancer research to reproduce our work. The detailed methods can be found on page 23 of the revised manuscript.

In relation to the concern about novelty, we acknowledge that it has been raised by another reviewer as well. We have addressed this concern in our reply to Reviewer #1, where we have provided a comprehensive explanation of the novelty of our findings. We have also taken this opportunity to further emphasize the novelty of our research in the revised Discussion on page 17.